# GloLakes: water storage dynamics for 27,000 lakes globally from 1984 to present derived from satellite altimetry and optical imaging

Jiawei Hou[1], Albert I.J.M. Van Dijk[1], Luigi J. Renzullo[1] and Pablo R. Larraondo[1]

[1]Fenner School of Environment & Society, Australian National University, Canberra, ACT, Australia

*Correspondence to*: Jiawei Hou (jiawei.hou@anu.edu.au)

**Abstract.** Measuring the spatiotemporal dynamics of lake and reservoir water storage is fundamental for assessing the influence of climate variability and anthropogenic activities on water quantity and quality. Previous studies estimated relative water volume changes for lakes where both satellite-derived extent and radar altimetry data are available. This approach is limited to only a few hundred lakes worldwide and cannot estimate absolute (i.e., total volume) water storage. We increased
the number of measured lakes by a factor of 300 by using high-resolution Landsat and Sentinel-2 optical remote sensing and ICESat-2 laser altimetry, in addition to radar altimetry from the Topex/Poseidon, Jason-1, -2 and -3, and Sentinel-3 and -6 instruments. Historical time series (1984-2020) of water storage could be derived for more than 170,000 lakes globally with a surface area of at least 1 km$^2$, representing 99% of the total volume of all water stored in lakes and reservoirs globally. Specifically, absolute lake volumes are estimated based on topographic characteristics and lake properties that can be observed
by remote sensing. In addition to that, we also generated relative lake volume changes solely based on satellite-derived heights and extents if both available. Within this data set, we investigated how many lakes can be measured in near real-time (2020-currrent) in basins worldwide. We developed an automated workflow for near real-time global lake monitoring of more than 27,000 lakes. The historical and near real-time lake storage dynamics data from 1984 to current are publicly available through https://doi.org/10.25914/K8ZF-6G46 (Hou et al., 2022) and a web-based data explorer www.globalwater.online.

## 1. Introduction

Lakes and reservoirs are a key component of the global water cycle through their contribution to land-atmosphere exchanges, river-floodplain dynamics and groundwater systems. Lakes also emit substantial quantities of carbon dioxide and methane into the atmosphere through biogeochemical processes (Bastviken et al., 2011; Raymond et al., 2013). The seasonality of natural lakes sustains ecosystems and biodiversity, whereas constructed reservoirs provide an often essential water supply to society
(Vörösmarty et al., 2010). Therefore, monitoring the quantity and quality of these surface water stores has important applications across a wide range of societal, environmental and economic areas.

Globally, surface water resources have become vulnerable to climate change and anthropogenic pressure (Vorosmarty et al., 2000). However, the processes, influences and consequences involved are still poorly understood in the absence of long-term

historical spatiotemporal dynamic information for lake dynamics. Shifts in seasonal cycles and extreme events are reported or predicted for many lakes due to climate change, which may worsen the already uneven distribution of water resources (Oki and Kanae, 2006; Wang et al., 2018). This stresses the need for global monitoring of lake dynamics, preferably in near real-time (NRT), e.g., a latency of 1~10 days.

Remote sensing approaches to global lake monitoring have benefitted from the substantial increase in Earth observation technologies over the last four decades (Papa et al., 2022). Remote sensing offers monitoring capability with coverage and consistency impossible to achieve with in situ networks (Alsdorf et al., 2007). Most natural lakes are ungauged, perhaps because they are generally more significant to ecosystems and biodiversity than for human activities and economic gain. Conversely, most large dam reservoirs are gauged, but the records are generally not publicly accessible. This lack of in situ
measurement impedes understanding the change and variability of lakes worldwide. Remote sensing provides a tool to tackle these issues and improve our incomplete knowledge about long-term changes in lakes at the local, regional and global scale.

Accurately locating lakes and reservoirs is the first step toward monitoring storage dynamics with remote sensing. Lehner and Döll (2004) used a range of data and digital maps to develop the Global Lakes and Wetlands Database (GLWD), which
delineated the boundaries of global lakes and reservoirs with a combined area of 2.7 million km$^2$. Based on GLWD and other regional and global data sources, Messager et al. (2016) developed the HydroLAKES database that provides detailed attribute information, such as shoreline length, size, and hydraulic residence times for 1.43 million lakes. The Global Water Bodies Database (GLOWABO) developed by Verpoorter et al. (2014) detected around 117 million lakes based on high-resolution satellite imagery. However, this dataset does not distinguish between lakes, rivers, floodplains and wetlands, unlike the
HydroLAKES database, which means that the number of lakes is overestimated. Lehner et al. (2011) compiled the storage capacity and characteristics of 6,862 dams and reservoirs in the Global Reservoir and Dam database (GRanD). By 2020, there were 58,713 dams registered by the International Commission on Large Dams (ICOLD), but most of them are still not georeferenced. This gap was addressed by the Global Georeferenced Database of Dams (GOODD), in which Mulligan et al. (2020) captured the locations of more than 38,000 dams from multiple satellite sources. The more recent Georeferenced global
Dams And Reservoirs (GeoDAR) dataset (Wang et al., 2022) not only provides the locations of 22,560 dams but also delineates the boundaries of 21,515 reservoirs around the world.

Radar altimetry, such as from the TOPEX/Poseidon, Jason-1/2/3, ENVISAT, ERS-1/2 and Sentinel-3/6 instruments, has proven useful in measuring water levels in lakes and reservoirs (Birkett, 1998; Da Silva et al., 2010; Frappart et al., 2006).
Global time series of radar altimetry-derived surface water elevation have been compiled in several places, including Hydroweb (Crétaux et al., 2011), the Database for Hydrological Time Series of Inland Waters (DAHITI) (Schwatke et al., 2015), and the Global Reservoirs and Lakes Monitor (GREALM) (Birkett et al., 2010). Based on the GREALM dataset, Kraemer et al. (2020) evaluated long-term trends in the water level of lakes globally, but their study was limited to around 200

lakes for which radar altimetry data were available. In contrast, Cooley et al. (2021) demonstrated the ability of ICESat-2 laser altimetry to measure water level variability for 227,386 lakes globally.

The spatial resolution of global satellite-derived surface water dynamics products improved significantly over the last two decades. Prigent et al. (2007) and Papa et al. (2010) developed a monthly and 25-km resolution surface water extent dataset based on a combination of passive (Special Sensor Microwave/Imager (SSM/I)) and active (European Remote Sensing (ERS)) microwave and optical remote sensing (Advanced Very High Resolution Radiometer (AVHRR)). This was refined to daily and 250 or 500-m resolution in the surface water dataset developed by Ji et al. (2018) and in the Global WaterPack (Klein et al., 2017). The 30-m resolution Global Surface Water Dataset (hereafter, GSWD (Landsat)) developed by Pekel et al. (2016) from imagery from the Landsat sensor series have been one of the most promising data sources to help understand long-term changes in surface water resources on Earth. Landsat archives have been the most popular data to investigate long-term changes and variability in lakes and reservoirs at either global or regional scale, benefiting from the four decades-long archives of images with global coverage and 30-m high resolution (Ogilvie et al., 2018; Sheng et al., 2016; Tao et al., 2015; Yao et al., 2019; Zhao and Gao, 2018).

Surface water height and extent are the two basic measurements needed to determine water storage change in lakes and reservoirs. Because of the shorter revisit times, daily or 8-day composite MODIS AQUA/TERRA products are less affected by cloud cover than 16-day Landsat observations, and the imagery is updated sufficiently rapidly to support near real-time (NRT) water monitoring. Some studies have used MODIS-derived water extent and altimetry data to estimate lake storage changes, such as in the Mackenzie Delta or South Asia, or worldwide (Gao et al., 2012; Normandin et al., 2018; Tortini et al., 2020; Zhang et al., 2014). However, the 500-m spatial resolution of MODIS fails to detect changes in smaller lakes, which are exponentially more numerous than large lakes. The 30-m resolution Landsat data, in combination with different altimetry sources, have been shown to be a better option for estimating water volume dynamics in lakes and reservoirs, especially those with relatively slowly changing extent (Busker et al., 2019; Duan and Bastiaanssen, 2013). Landsat satellite series can provide historical observations back to the 1980s. Finally, while MODIS- or Landsat-derived water extent and altimetry data have demonstrated the capability to estimate changes in lake water storage, they cannot measure lake depth and, therefore, cannot provide absolute water storage volume without using bathymetric data. Several studies (e.g., Avisse et al., 2017; Bonnema et al., 2016; Vu et al., 2022) used a digital elevation model (DEM) to derive height-area curves and estimate absolute water volume, but the success of this approach depends on the volume of water present at the time of DEM data acquisition (e.g., February 2000 for the Shuttle Radar Topography Mission DEM data often used). To circumvent this, Messager et al. (2016) used the surrounding terrain data (i.e., slope derived from the DEM), while Khazaei et al. (2022) used geophysical characteristics and hypothetical idealised geometry (i.e., cone, box, triangular prism, and ellipsoid) to estimate water depth for lakes in the absence of bathymetry. Despite those approaches, there are currently no data products that provide absolute lake volume estimates at the global scale.

The objectives of this study were to (1) derive nearly four decades of data on relative and absolute water volume measurements for lakes worldwide and (2) enable a global lake monitoring capability. To develop this, we first estimated water body extents between 1984-2020 from Landsat-derived surface water maps (i.e., GSWD (Landsat)) for 170,957 lakes with a surface area of at least 1 km$^2$. We applied the gap-filling algorithm (Hou et al., 2022) in contaminated GSWD (Landsat) surface water images to restore missing data, thus improving the total number of usable images to derive lake area time series. Second, we estimated absolute water storage dynamics from 1984-2020 for each lake whose water area and its surrounding slope measurements are available using a geostatistical model (Messager et al., 2016). Third, we extended lake volume data beyond 2020 to provide near real-time or at least up-to-date information (from hereon referred to as 'NRT' for brevity). We considered a range of alternative satellite data sources (including Sentinel-2, Jason-3, Sentinel-3 and -6, and ICESat-2) with monitoring abilities to derive NRT absolute lake storage. We examined where and for how many lakes NRT storage can be estimated using different combinations of these remote sensing data in each basin worldwide. Fourth, we extended historical absolute water storage estimates to NRT monitoring using the volume-height relationship if radar or lidar altimetry data available after 2020. Where only NRT lake water area observations (e.g., from Sentinel-2) were available, we converted lake area to storage estimates using a geostatistical model. As these absolute lake storage products are affected by the bias errors of lake depth estimates from the geostatistical model, we also provided relative lake storage estimates for lakes observed simultaneously by optical imagers and altimetry. Overall, this made it possible to monitor more than 27,000 lakes and reservoirs worldwide. The underlying strategy in developing this global lake monitoring system was to evaluate a selection of readily available, validated, and frequently updated satellite data sources, explore the relative advantage of each selected source, and combine them to complement their respective weaknesses.

## 2. Data and method

### 2.1 Data

#### 2.1.1 Surface water extent

The Joint Research Centre's Global Surface Water Dataset (GSWD (Landsat)) provides the spatial and temporal distribution of surface water and their statistics at a global scale over the last 37 years. Open water areas larger than ca. 30 m × 30 m were detected by an expert system using Landsat-5 Thematic Mapper (TM), Landsat-7 Enhanced Thematic Mapper-plus (ETM+ ) and Landsat-8 Operational Land Imager (OLI) images between 1984 and 2020 (Pekel et al., 2016). The omission errors of water mapping from Landsat-5, -7 and -8 are less than 5%, while the commission errors are less than 1%. The monthly water history and monthly recurrence products from GSWD were used here. The monthly water history product provides monthly water mapping from March 1984 to December 2020. Each pixel was classified as open water, land, and non-valid observation. The monthly recurrence product comprises 12 datasets, one for each month (from January to December). Each shows the frequency of inundation in each pixel as a percentage of the number of times water is detected over the total number of clear

observations in the full-time series. The BLUEDOT water observatory (hereafter BLUEDOT (Sentinel-2); https://www.blue-dot-observatory.com/aboutwaterobservatory) provides five-day NRT measurements of surface water extent from 2015 to the present for 8,837 lakes globally. The surface water map is derived using normalised difference water index (NDWI) from clear-day optical Sentinel-2 imagery. We used these data to complement our Landsat data.

### 2.1.2 Surface water height

The Advanced Topographic Laser Altimeter System (ATLAS) onboard the Ice, Cloud and land Elevation Satellite-2 (ICESat-2) launched by NASA in 2018 is designed to measure the elevation of ice sheets, oceans, lakes and vegetation with a 91-day repeat cycle. ATLAS/ICESat-2 determines elevation by measuring the return time of a laser pulse between the satellite and the Earth's surface. The six laser beams from ICESat-2 allow it to cover more ground coverage of the Earth's surface than its predecessor (ICESat-1). We used the ATLAS/ICESat-2 L3A Along-Track Inland Surface Water Data, Version 5 (ATL13)

(Jasinski and Ondrusek, 2021). Unfortunately, updated ATL13 data are not released until 30-45 days after new observations are obtained, which does not suit NRT monitoring. Therefore, we also used ATLAS/ICESat-2 L3A Along Track Inland Surface Water Data Quick Look, Version 5 (ATL13QL), available within three days of new observations. ATL13 and ATL13QL apply the same algorithms to derive surface water height measurements for inland water bodies, including rivers, lakes, reservoirs and coastal water. The approach is described by Jasinski and Ondrusek (2021), but in brief, the procedure: 1) identifies ATLAS

beams that intersect inland water body shape masks; 2) collects photons in short segments (~100 m) for calculating water height and in longer segments (1~3 km) for estimating and removing subsurface backscatter; 3) applies the physical and statistical modelling to derive inland water heights. Cooley et al. (2021) used ATLAS/ICESat-2 L3A Land and Vegetation Height (ATL08) to derive lake height time series and found the mean absolute error (MAE) between USGS gauge data and ICESat-2 height measurements is 0.14 m. Unlike ATL08, the ATL13 used here was designed to measure water height

variations in rivers and lakes as it considers physical processes of light propagation in open water bodies. An error of 6.1 cm per 100 inland water photons has been reported (Jasinski and Ondrusek, 2021). We collected both ATL13 and ATL13QL water surface height data from any of the six beams within individual lake boundaries from HydroLAKES and derived water height for each lake observed by ICESat-2 between 2018 and the present. ATL13QL has larger uncertainties (~100 m) in geolocation than ATL13 (~5 m). As a result, segment heights from ATL13QL are 2.7 m, with a standard deviation of ~ 7 m,

lower than those from ATL13. According to the user guide (Jasinski and Ondrusek, 2021), 2.7 m can be added to ATL13QL before merging these two products, and the differences between them have little impact on measuring relative heights. Each lake will be revisited by ICESat-2 around every 91 days. This means our ICESat-2-based storage product is limited to providing up-to-date information on seasonal variation rather than short-term dynamics. However, it has the benefit of providing observations with a rapid turnaround.

The Global Reservoirs and Lakes Monitor (GREALM) provides NRT surface water height dynamics for around 500 lakes globally using a combination of different satellite radar altimetry (Birkett et al., 2010). It provides two ongoing products (1)

10-day NRT water heights from 1993-present derived from Topex/Poseidon, Jason 1/2/3 and Sentinel-6, and (2) 27-day NRT water heights from 2016-present using Sentinel-3A and B. Satellite radar altimetry determines the surface height by emitting microwave pulses towards the Earth's surface and measuring the travel time between pulse emission and echo reception. The accuracy of these radar altimetry-derived water heights varies, mainly depending on the roughness and extent of the water body, but is typically within a few cm.

### 2.1.3 Geostatistical model

The HydroLAKES database provides the boundary outlines of more than 1.4 million individual lakes globally with a surface area above 0.1 km$^2$ (Messager et al., 2016). This database is compiled from several global and regional lake and reservoir datasets derived using topographic maps, optical remote sensing imagery composites and radar instruments. HydroLAKES also comprises a geostatistical model with parameters to predict average water depths and volumes based on surrounding topography information for lakes with a surface extent between 0.1 km$^2$ and 500 km$^2$. The geostatistical model (Messager et al., 2016) is described as follows:

$$\text{Log}_{10}(D) = C_1 + C_2 \times \text{Log}_{10}(A) + C_3 \times \text{Log}_{10}(S_{100}) + s^2 \qquad (1)$$

where $D$ is the predicted mean depth (m), $A$ the observed surface area of the lake (km$^2$), $S_{100}$ the average slope (derived from DEM) within a 100-m buffer around the lake, $C_1$, $C_2$ and $C_3$ coefficients estimated from fitting the model using global lake data for different lakes sizes (i.e., 0.1-1 km$^2$, 1-10 km$^2$, 10-100 km$^2$, 100-500 km$^2$), and $s^2$ the residual variance. The fundamental assumption is that one can extrapolate the slope around the lake towards the centre of the lake to estimate lake depth. The traditional approach uses a DEM within the lake to derive area-height curves but cannot retrieve true land surface elevation below water. In comparison, the model used here considers the DEM outside lake rather than inside it and hence is not affected by water inundation at the time of DEM acquisition. Messager et al. (2016) reported that the symmetric mean absolute percent error (SMAPE) between predicted and reference volume is 48.8% without significant bias in volumes for the majority of lakes around the world, with the exception of Finland, Sweden and northwestern Russia, the European Alps, and the Andes. We compared this approach with alternative approaches. Khazaei et al. (2022) developed a statistical model, GLOBahty, relating lake depth with surface water area, elevation, volume, shoreline length, and watershed area. They demonstrated that the performance of this statistical model is better than previous approaches to calculating lake depth by assuming the lake shape fits one of four geometries: box, cone, triangular prism, and ellipsoid (e.g., Yigzaw et al., 2018). We compared both models to in situ data. The resulting SMAPE error for the GLOBahty method was 70.5%, and therefore not better than the geostatistical method. This result informed our choice to use the simpler geostatistical method in favour of the GLOBathy dataset in this study. The uncertainties from the geostatistical model are mainly from the observed surface area and the DEM used to derive slope. The omission and commission errors of surface water mapping from GSWD used in this study are 5% and 1%, respectively. EarthEnv-DEM90 was used in the geostatistical model to derive slope and its vertical accuracy

is around 10 m (Robinson et al., 2014). We assessed how these errors propagate into water storage estimation in the validation analysis.

## 2.2 Method

### 2.2.1 Global historical lake volume estimation

We estimated water surface extent changes of GSWD (Landsat) water bodies, where the shoreline polygons of lakes or reservoirs were delineated with HydroLAKES, as follows. First, we overlaid each lake boundary polygon from HydroLAKES on GSWD (Landsat). Second, we introduced a 500-m buffer around the lake to ensure the maximum water extent possible was enveloped. Third, for each lake, we calculated the number of wet pixels within the lake boundary polygon from the GSWD (Landsat) monthly water history product from 1984 to 2020. Lake water extent was then estimated by multiplying the total

number of wet pixels by the grid pixel area. During this process, we also calculated the contamination ratio of each image as the ratio of non-valid pixel values over all pixel values within the lake boundaries. Water extent was not estimated for lakes smaller than 1 km$^2$ due to the limited spatial resolution of Landsat. Finally, monthly lake extent time series were produced for 170,957 lakes worldwide.

Landsat images are affected by suboptimal observation conditions (e.g., cloud and cloud shadows) and data acquisition (e.g., swath edges, the Landsat-7 Scan Line Corrector failure) and limited archiving of acquired imagery. These factors reduced the total number of effective Landsat images available over the 37 years. Previous studies used images with a contamination ratio below a certain threshold (e.g., 5%) or aggregated the imagery to seasonal, annual or five-year averages (e.g., Shugar et al., 2020; Yang et al., 2020). However, low-frequency time series can miss important monthly, seasonal and interannual changes

in surface water extent. Alternatively, higher temporal frequency satellite data such as MODIS may be used to fill gaps in the Landsat imagery (e.g., Li et al., 2021). However, this is only possible for the MODIS era, i.e., after 2000, and the spatial resolution of MODIS poses an additional constraint on the size of the lake that can be monitored.

Here, we applied a simple and effective gap-filling approach we published previously (Hou et al., 2022) to recover missing

data in partial Landsat water mapping to boost the total number of usable images. Unlike our earlier publication, however, we here used a monthly recurrence map rather than a multi-year average recurrence map. This made it possible to restore the partially missing maps for the specific month considered, making it more sensitive to seasonal changes. In summary, the monthly GSWD (Landsat) recurrence product was used to restore missing data in any given month (January to December), considering the seasonal changes in surface water extent. For example, the contaminated surface water map derived from

GSWD (Landsat) in January 2022 was restored using the January recurrence data. This was achieved by, first, matching the GSWD (Landsat) extent mapping for different recurrence frequencies in the corresponding month to the available parts of each contaminated image. The differential evolution method (Storn and Price, 1997) was used to find the best-fit frequency,

minimising the difference between the recurrence mapping and monthly water mapping. Subsequently, the mapping at the best-fit recurrence frequency was used to reconstruct missing data. The algorithm was implemented for all images with contamination ratios ranging from 5% up to 70%. The efficacy of this approach was evaluated and confirmed in a previous publication (Hou et al., 2022).

For each lake, mean lake water depth dynamics were calculated based on the estimated water extent time series using the mentioned geostatistical model of Messager et al. (2016). The predicted mean water depths were bias-corrected by the residual variance of its corresponding empirical equations (Messager et al., 2016). Lake water volumes (V in gigalitre (GL), millions of cubic meters (MCM) or $10^6\,\mathrm{m}^3$) were estimated by the bias-corrected predicted water depths multiplied by water extents. Ultimately, we estimated monthly water storage dynamics from 1984-2020 for 170,611 lakes globally.

### 2.2.2 Global near real-time lake volume estimation

To provide routine and low-latency measurements of lake water storage, we obtained NRT satellite-derived water heights and extents from different monitoring satellite sources, including ICESat-2, GREALM and BLUEDOT (Sentinel-2) water observatory (Table 1). We selected all estimates for lakes whose volume dynamics from 1984–2020 were derived from GSWD (Landsat) with the geostatistical model. We investigated where and for how many lakes both historical and NRT storages can be estimated reliably around the world. We assumed that if satellite-derived extents and levels (e.g., GSWD (Landsat) and ICESat-2) or two satellite sources (i.e., GSWD (Landsat) and BLUEDOT (Sentinel-2)) show consistency in lake changes over time, we are able to ensure the reliability of these satellite measurements to derive lake volume estimates. We examined pairwise relationships (i.e., $A_{his}$-$H_{nrt}$ or $A_{his}$-$A_{nrt}$) of overlapping monthly time series between historical area ($A_{his}$) and NRT height ($H_{nrt}$) or area ($A_{nrt}$). We only extended historical volume estimation to NRT monitoring when there was a statistically significant ($p<0.05$) correlation. Specifically, we calculated the significant Pearson correlation threshold ($R_t$) using a t-test allowing for sample size $N$ (i.e., the number of data pairs). If the Pearson correlation ($R$) between $A_{his}$-$H_{nrt}$ or $A_{his}$-$A_{nrt}$ exceeded $R_t$ we used $H_{nrt}$ or $A_{nrt}$, respectively, to estimate lake storage dynamics after 2020.

Depending on some of the features of the satellite data sources (BLUEDOT (Sentinel-2), ICESat-2, GREALM), we used different approaches to estimate absolute NRT storage. If both historical and NRT sources (i.e., GSWD (Landsat) and BLUEDOT (Sentinel-2)) only observed lake area changes, we converted lake area to storage using the geostatistical model (Eq. 1) and merged them to derive storage time series from 1984-present. If NRT satellite sources (i.e., ICESat-2 or GREALM) observed lake water height changes, we developed the relationship between historical volume ($V_{his}$) and NRT height ($H_{nrt}$) for the overlapping months and used that $V_{his}$-$H_{nrt}$, if significantly correlated, to extend historical volume estimation to NRT monitoring using only satellite-derived $H_{nrt}$. As the relationship is not necessarily linear, NRT lake storage was estimated using cumulative distribution function (CDF) matching. Unlike the conventional approach to build $V$-$A$-$H$ curves, which typically requires the selection and fitting of empirical equations (e.g., linear or higher-degree polynomial equations), CDF matching

takes a simpler route. It entails the development of a monotonic cumulative distribution function for overlapping height ($H_{nrt}$) and volume ($V_{his}$) data. The distribution allows for the ranking of values, showing their relative positions in the dataset. When a new height value ($H_{nrt}$) is to be retrieved, CDF matching uses the pre-established cumulative distribution functions and their associated rankings. By comparing the rank of the new value with those in the distributions, the corresponding volume value ($V_{nrt}$) can be determined. Overall, we derived absolute water storage dynamics from 1984 onwards for 24,865 lakes using a geostatistical-model and GSWD (Landsat) with ICESat-2, 129 lakes with GREALM, and 4,054 lakes with BLUEDOT (Sentinel-2).

Our main objective was to estimate NRT absolute water storage dynamics for lakes worldwide as much as possible. However, we also measured relative storage where radar or laser-derived surface water heights, as well as optical imaging-derived surface water extent, were available (i.e., GSWD (Landsat) with ICESat-2 or GREALM, and BLUEDOT (Sentinel-2) and ICESat-2). Unlike absolute storage products, relative storage products are unaffected by any bias from the geostatistical model. We produced alternative storage datasets to provide options to users requiring different bias tolerances or different applications. For the relative storage products, we calculated historical storage changes for lakes where the *A-H* relationship was significantly correlated as follows (Crétaux et al., 2016):

$$\Delta V = \frac{(H_t - H_{t-1}) \times (A_t + A_{t-1} + \sqrt{A_t \times A_{t-1}})}{3} \qquad (2)$$

where $\Delta V$ is storage change between two consecutive measurements; $H_t$ and $H_{t-1}$ are altimetry-derived surface water heights at time *t* and *t-1*, respectively; $A_t$ and $A_{t-1}$ are optical remote sensing derived surface water extents at time *t* and *t-1*, respectively. Where there are only *H* data available in the NRT period, we estimated corresponding *A* using CDF matching and calculated relative storage change time series using Eq (2). As Sentinel-2 and ICESat-2 can measure NRT surface water extent and level, respectively, we used them to derive relative storage changes from 2018-present. This product is free from any errors in the geostatistical model and in the *A-H* or *V-H* relationships used in the other products. Overall, we measured relative lake storage dynamics from 2018 onwards for 24,990 lakes using ICESat-2 and GSWD (Landsat), for 2,740 lakes using ICESat-2 and BLUEDOT (Sentinel-2), and from 1993 onwards for 227 lakes using GREALM and GSWD (Landsat).

**Table 1** Sources of satellite data for lake storage monitoring using observations overlapping with the GSWD (Landsat) surface water extent maps.

| Dataset | I | II | III |
| --- | --- | --- | --- |
| Source | ICESat-2 | GREALM | BLUEDOT (Sentinel-2) |
| Period | 2018-current | 1993-current/2016-current | 2015-current |
| Type | laser altimetry | radar altimetry | optical |
| Variable | height | height | extent |
| Temporal resolution | 91 days | 10-days/27-days | 5-days |
| Overlapping with GSWD (Landsat) (number) | 101,983 | 347 | 5,948 |

## 3. Results and Discussion

### 3.1 Lake area estimation benchmarking

A large portion of Landsat data was missing due to cloud, cloud shadow and the Landsat-7 Scan Line Corrector (SLC) failure. Left unmitigated, this issue would have limited the observations per year or an underestimation of surface water area even if using images with little contamination. Cloud cover was the most common issue, but water extent could be reconstructed for contamination ratios of 58% and higher (Fig. 1 and Fig. S1). The SLC failure resulted in missing data of around 22% after 2003 (Chen et al., 2011). Missing data could also be caused by a combination of cloud cover, SLC failure and swath edges, for example (Fig. 1 and Fig. S1). The reconstructed water extent maps in the rightmost column of Fig. 1 and Fig. S1 demonstrate that these missing data scenarios could be usefully restored, no matter what types of contamination issues, what kinds of lake shapes, or how much (between 5-70%) of contamination ratios. By applying this step, we strongly increased the number of usable images.

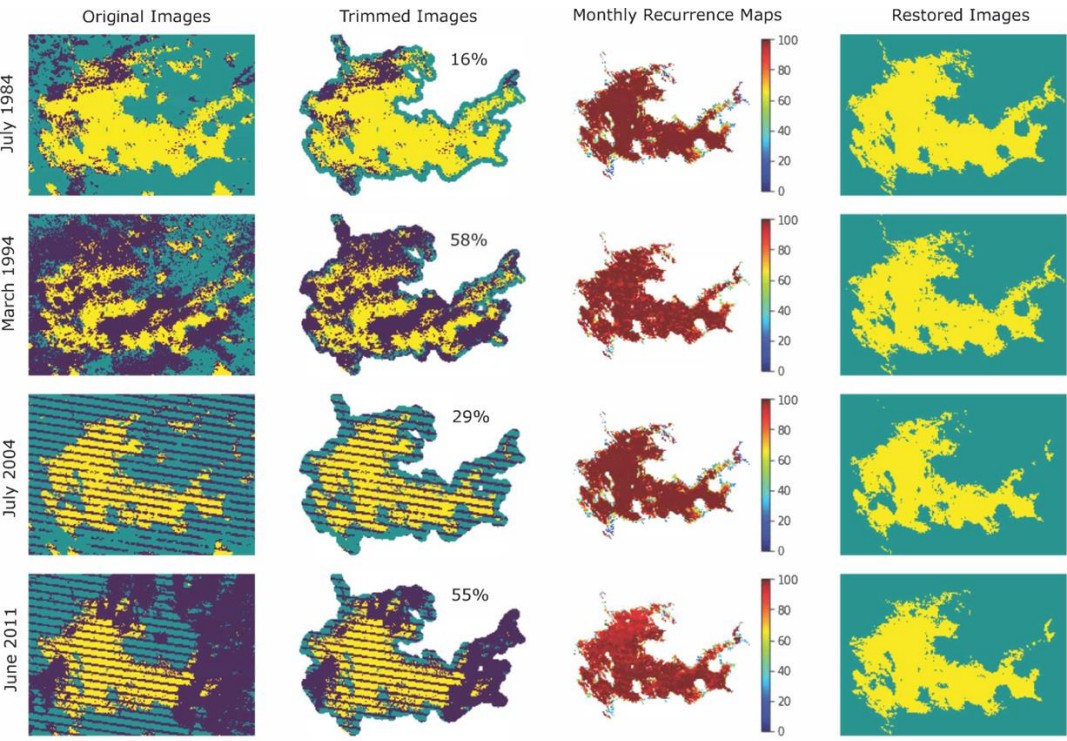

**Figure 1.** Examples of the performance of the image gap-filling algorithm in Lake Rossignol, Canada. (First column: historical water maps from GSWD (Landsat) (yellow: water; green: land; blue: no data; image contaminated ratio from top to bottom: 16%, 58%, 29% and 55%); second column: historical water maps trimmed by the lake boundary from HydroLAKES with a 500 m buffer (number: contamination ratio); third column: water recurrence (0~100%) maps at the specific month; fourth column: restored water maps)

Our lake area data have 5318 lakes in common with the data of Zhao and Gao (2018) and 11,101 lakes with Donchyts et al. (2022) with average areas from 0.1 km$^2$ to 1000 km$^2$. Zhao and Gao (2018) used the same Landsat data source (i.e., GSWD)

and the same lake boundary delineation (GRanD included in HydroLAKES) but a different gap-filling approach to derive lake area. Nonetheless, the mean lake area between the two products scatters closely around a 1:1 relationship (Fig. 2a). We also calculated correlation and bias in lake area time series for the common period 1984–2018 for each lake. The median R and SMAPE were 0.91 and 3.6%, respectively. This suggests that our gap-filling algorithm produces results overall similar to those of Zhao and Gao (2018). Donchyts et al. (2022) used a different Landsat data source (i.e., derived spectral water index), lake

boundary delineation, and gap-filling algorithm to derive lake area time series. Despite these differences, mean lake area values still cluster fairly closely around the 1:1 relationship, especially for lakes greater than 1 km$^2$ (Fig. 2b). Larger biases exist for some lakes smaller than 1 km$^2$. This was caused mainly by different definitions of lake boundaries between HydroLAKES and Donchyts et al. (2022). The median R and SMAPE in lake area time series from 1984 to 2020 for 11,101 lakes between our product and Donchyts et al. (2022) were 0.76 and 9.7%, once again lower mainly due to the differences for small lakes.

Although there are limitations and challenges when using remote sensing to precisely estimate area change for each lake, especially in global studies, this benchmarking analysis has ensured that our lake extent estimation attains a level of proficiency comparable to other published datasets.

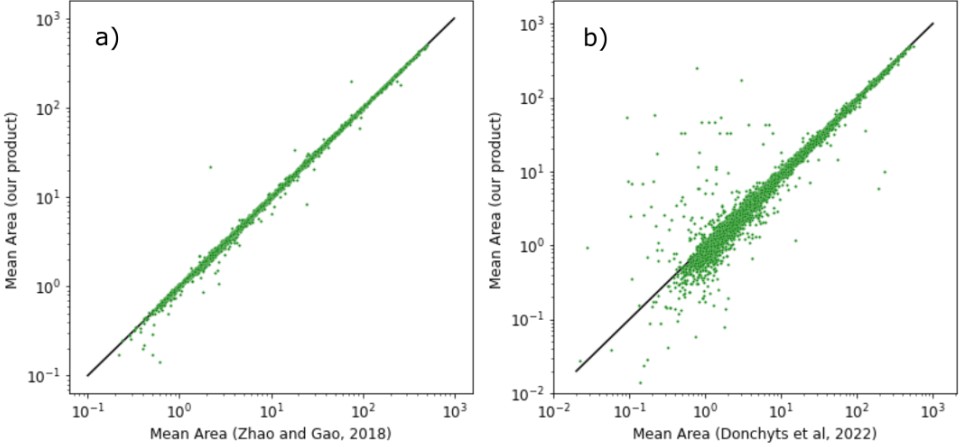

**Figure 2** Scatterplots of mean lake area from our product vs. two other published datasets (black line: 1:1 relationship).

To further assess our lake extent estimates, we manually derived lake area based on bands 8 (near infrared), 4 (red) and 3 (green) from high-resolution Sentinel-2 imagery using Google Earth Engine and compared it against lake area derived in this

study (Figure S2 and Table S1). We selected two Sentinel-2 images without cloud cover - one with a relatively small extent and another with a relatively large extent - for each of 20 lakes chosen to represent different continents and climates and with varying surface water color, lake shape and size and surrounding land cover. We delineated the lake area using Sentinel-2 surface reflectances in the three bands and visually benchmarked it with a false-color image. The Sentinel-2 derived lake area was then compared to GSWD (Landsat) derived lake area. The results show that the average difference of lake area estimates derived between GSWD (Landsat) and Sentinel-2 for all 40 pairs is 2.62% (Table S1). The estimated lake area differences do not vary systematically with geographical location, surrounding land cover or lake shape and size. Most importantly, the area estimates for both small and large extents exhibited very strong consistency between the two satellite sources (Figure S2). This analysis supports that GSWD (Landsat) can effectively capture changes in lake area. Comparatively large biases (from 3-10%) are observed exclusively in lakes with surrounding ephemeral disconnected surface water bodies or emerging islands within the lake during dry periods, such as Lake Pozuelos in Argentina, Lake Baia Grande in Brazil, and Lake Aksehir in Turkey (Figure S2 and Table S1).

## 3.2 Feasibility of near real-time lake monitoring

The primary objective behind generating the NRT lake data is to evaluate present-day conditions in lakes and dams in their historical context. This becomes takes on ever greater importance in an era of rapid climate change. Of particular significance is the storage volume, as it enables the aggregation of numerous water bodies within a single system. Such an aggregation offers a more comprehensive and insightful perspective on the hydrological status of catchments or river systems, such as provided by initiatives like the Global Water Monitor (www.globalwater.online). Knowledge of combined stored volumes are also needed to understand concepts such as catchment water equilibrium and to interpret GRACE terrestrial water storage estimates for example, enhancing our understanding of global water availability.

We estimated monthly lake volume dynamics for 170,611 lakes during 1984–2020 (Fig. 3a). Most lakes are in the northern hemisphere's high latitudes, especially in North America. We only considered lakes with an area greater than 1 km$^2$ given the resolution of the Landsat imagery. There are more than a million lakes with areas between 0.1–1 km$^2$ that remain unmeasured, but the combined storage of these small lakes only accounts for 1% of global total lake water storage, according to the HydroLAKES dataset. Overall, a large portion of global lake volume for the period 1984-2020 was estimated. Here we focus on estimating both historical and NRT lake dynamics. NRT lake water storage was only estimated if *A-H-V* relationships were sufficiently strong, even though we produced historical water storage change for more than 170,000 lakes. This also ensures that satellite-derived extent and level observations are consistent in characterising lake change before producing lake storage time series from 1984 to the present.

As the Landsat, Sentinel-2 and ICESat-2 have comprehensive coverage of global lakes, we investigated if any two of them (optical + altimetry or two optical) can be used to monitor global lakes and how much of the lakes in each basin storage can

be measured in each case. This should provide valuable information for the newly launched Surface Water and Ocean Topography (SWOT) mission that aims to monitor storage changes in global lakes by measuring both extent and level on a single satellite platform. The HydroBASINS database delineates watershed boundaries worldwide at different basin or catchments scales (Lehner and Grill, 2013). The catchment boundaries in the Pfafstetter level-3 product from HydroBASINS were used as the spatial basin units for analysis. If remote sensing measurements from two sources are significantly correlated for a particular lake, we considered that this lake can be monitored by Earth observation. We followed three complementary approaches: (1) geostatistical model + extent (GSWD (Landsat) + geostatistical model + BLUEDOT (Sentinel-2)), (2) geostatistical model + height (GSWD (Landsat) + geostatistical model + $V$-$H$ relationship + ICESat-2) and (3) extent + height (BLUEDOT (Sentinel-2) + ICESat-2). Approach (3) would likely be the most reliable as storage is directly measured by extent and level from remote sensing. However, approaches (1) and (2) make it possible to estimate absolute storage changes. GSWD (Landsat) and ICESat-2 together could measure lake changes in nearly all (i.e., 234 out of 292) river basins worldwide (Fig. 3b). Extents derived from GSWD (Landsat) data and levels derived from ICESat-2 exhibit a significant correlation in over 25% of lakes measured by both in each of 145 river basins. The correlation pattern is evenly distributed across the continents, except for Antarctica and high northern latitudes as storage changes in many lakes in these cold regions are dominated by level changes or there may be large uncertainties in surface water maps due to the influence of frozen water surfaces. BLUEDOT (Sentinel-2) and ICESat-2 cover 122 basins globally (Fig. 3d). There are 58 basins in each of which over half of the lakes measured by both satellite sources can be monitored by that method, mainly located in the USA, southeastern South America, the Mediterranean, southern Africa, southern Asia, and Australia. GSWD (Landsat) and BLUEDOT (Sentinel-2) both measure surface water extent and show strong time series consistency in 63 out of 124 basins. Three-quarters of lakes in each of these 63 basins have a significant $A$-$H$ relationship (Fig. 3c) with a distribution pattern similar to that of BLUEDOT (Sentinel-2)/ICESat-2.

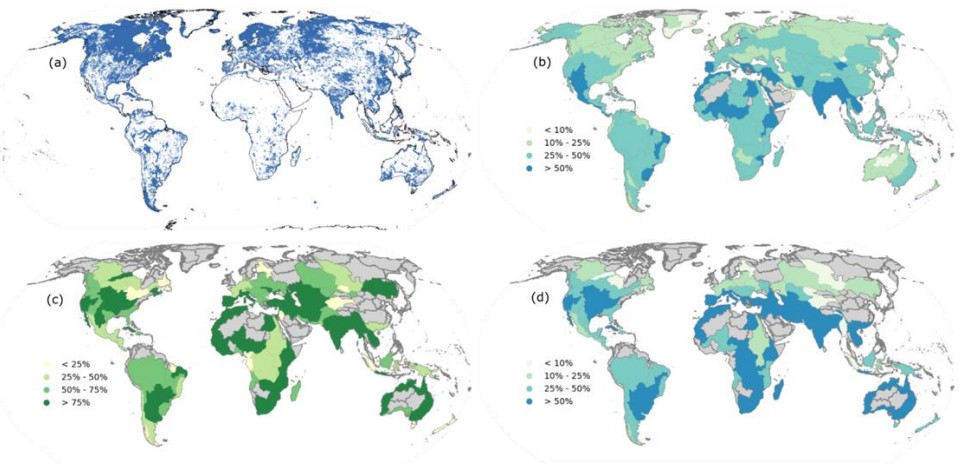

**Figure 3** The locations of 170,611 lakes whose storage dynamics for the period of 1984-2020 were estimated in this study (a), the percentages of lakes (refer to the total number of lakes in each basin according to Fig 3a) whose changes are consistently measured by both

GSWD (Landsat) and ICESat-2 (b), GSWD (Landsat) and BLUEDOT (Sentinel-2) (c), and BLUEDOT (Sentinel-2) and ICESat-2 (d) in each basin.


### 3.3 Validation of historical lake storage estimates

There is a general lack of independent and publicly available lake and reservoir storage data to validate our results, and indeed the lack of on-ground data motivated our study. Nonetheless, we were able to validate absolute lake volume estimates against in situ measurements for 494 lakes in Australia, South Africa, India, Spain and the USA, respectively available from the
Australian Bureau of Meteorology, Donchyts et al. (2022), United States Bureau of Reclamation and United States Army Corps of Engineers. We evaluated the accuracy of lake volume time series by Pearson correlation ($R$) and the symmetric mean absolute percent error:

$$SMAPE = 100 \times \frac{1}{N} \sum \frac{|observed\ volume - predicted\ volume|}{(observed\ volume + predicted\ volume)/2} \quad (3)$$

The results (Fig. 4) suggest a median $R$ between reported lake volumes and our estimates of 0.91 and a SMAPE of 38%. This
SMAPE is similar to the 48.8% reported by Messager et al. (2016). 82% of validated lakes have $R$ above 0.7 and SMAPE mainly ranges between 5–50% (Fig. 4). Comparisons for selected lakes are shown in Fig.5, representing different sizes of lakes and different SMAPE errors. It appears that uncertainties from remote sensing and the DEM do not meaningfully affect absolute lake volume estimates (Fig. 5). The lakes for which storage was validated had average volumes between 1 and 10,000 GL (Fig. 6). The comparison between GloLakes and in situ data show that the bias error (38%) is consistent for different lake
sizes (Fig. 6).

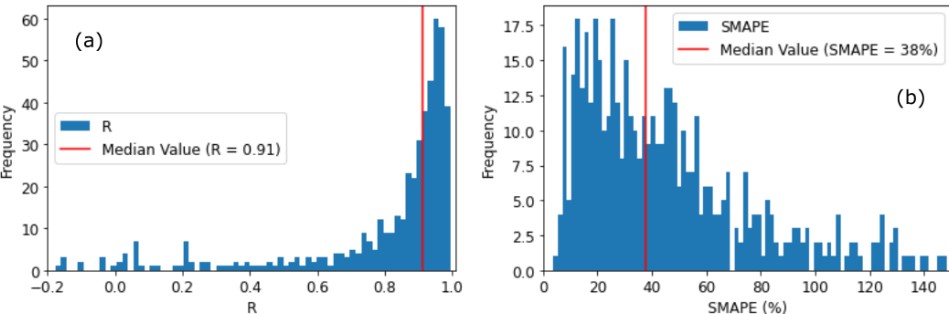

**Figure 4** The distribution of R and SMAPE of absolute water storage validation results against in situ data for 494 lakes (vertical red line: the median value).


![Figure 5 time series panels](CHAFFEY DAM; SPLIT ROCK DAM; Cairn Curran Reservoir; Pike_Ck Glenlyon HW; Greens Lake; GLENLYON DAM; Lake Bellfield; Somerset; Lake Eucumbene RL; Bjelke-Petersen Dam; Nepean; Lake Paluma)

**Figure 5** Absolute lake water storage (unit: GL) time series from GloLakes against in situ (observed) data for selected lakes (blue line: observed data; red line: historical storage estimates; black shade: error bars of historical storage estimates).


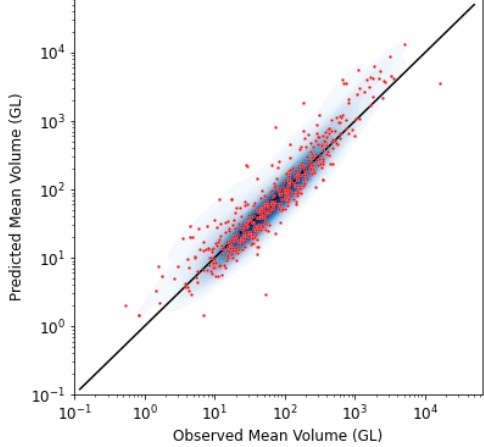

**Figure 6** Scatterplots of predicted mean lake volume vs. observed mean volume (red dot: mean lake volume; black line: 1:1 relationship; background blue colours indicate data density).

For many applications, the relative agreement (e.g., $R$) may be more relevant than the absolute error, and the latter can be
affected by several factors. For example, the delineation of lake shorelines from HydroLAKES can be different from those
used for in situ measurement, as the spatial definition to distinguish the lake from connected rivers or wetlands can sometimes
be ambiguous. In such cases, predicted volumes are not directly comparable to in situ data, and there can be systematic over-
or underestimation. Secondly, lake volume cannot be estimated with high accuracy unless detailed lake bathymetry data is
available, but this is beyond current satellite remote sensing abilities. We used a geostatistical model to estimate lake depth.
The accuracy of that approach depends on the quality of the DEM and the degree to which the relationship between the
topography of the lakebed and surrounding slopes conforms to the geostatistical model assumptions.

The relative volume estimates are generally more reliable, as demonstrated by the correlation values. We compared our relative
lake storage dynamics data against the global surface water storage change time series for 1992–2018 produced by Tortini et
al. (2020) for 104 common lakes. The data was downloaded from the Physical Oceanography Distributed Active Archive
Center (PODAAC) of NASA's Jet Propulsion Laboratory. The two show strong agreement (Fig. 7) with a median $R$ of 0.94
and 95% of $R$ values above 0.7. Selected time series are compared in Fig. 8 and show that both datasets are most limited by
the temporal frequency of image acquisition.

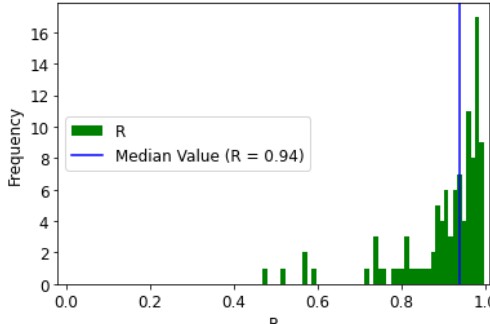

**Figure 7** The distribution of R of relative water storage benchmarking results against Tortini et al. (2022) for 104 lakes (vertical blue line:
the median value).

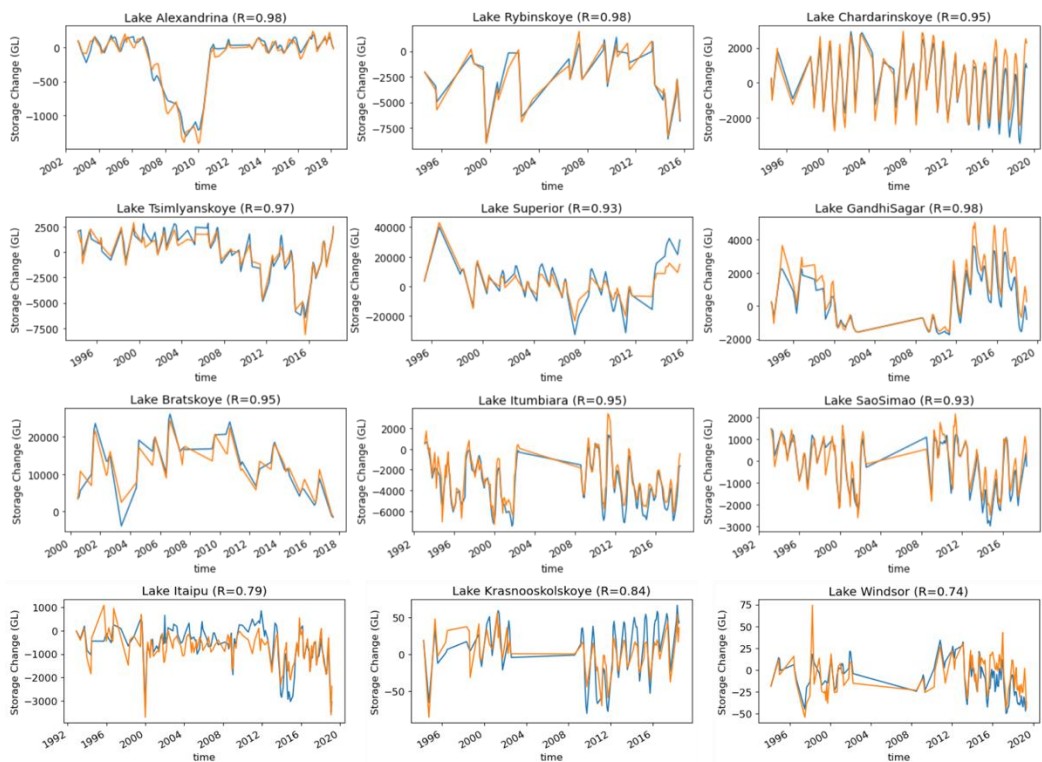

**Figure 8** Relative lake water storage (unit: GL) time series from GloLakes (blue line) against Tortini et al. (2022) (brown line) for selected lakes.


### 3.4 Validation of NRT lake storage estimates

To our knowledge, we produced the first four decades-long data set of relative and absolute lake storage dynamics from 1984-
present for all HydroLAKES registered lakes with an area exceeding 1 km². Previous studies have focused on measuring relative lake storage dynamics using remote sensing. For example, Busker et al. (2019) used Landsat imagery and radar altimetry to estimate lake volume variations for 137 lakes. Similarly, Tortini et al. (2020) produced a storage change dataset for 347 lakes but using MODIS instead of Landsat imagery. Neither study offered an NRT monitoring capacity. They were also limited to a few hundred lakes due to the sparse track of radar altimetry. With denser coverage, the ICESat-2 laser altimetry
improves the number of measured lakes by a factor of more than hundred. Based on this, we were able to monitor a wide range of lakes at global scale.

To validate satellite-derived lake heights used to estimate NRT storages in this study, we obtained in situ lake level measurements from the Australian Bureau of Meteorology (http://www.bom.gov.au/water/index.shtml) and Environment
Canada (https://wateroffice.ec.gc.ca/). In total, we identified 96 lakes in Australia (including 83 reservoirs) and 54 lakes in

Canada (including 23 reservoirs) with in situ level data corresponding to our study. Following the same validation approach as Cooley et al. (2021), we compared temporal changes in lake heights between in situ data and altimetry data. The results show strong agreement between ICESat-2 and in situ measurements, with a mean absolute error (MAE) of 0.23 m and a mean Pearson correlation (R) of 0.99 in Australia and 0.19 m, and 0.97 in Canada (Figure 9).

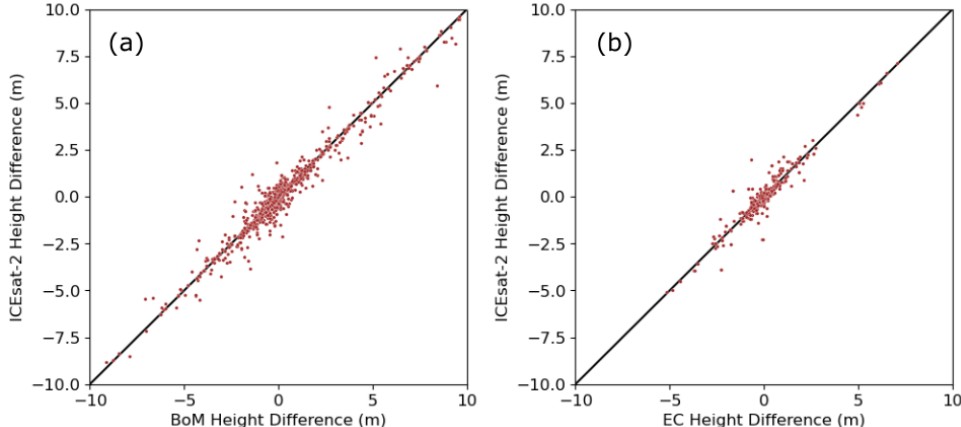

**Figure 9** Evaluation of ICESat-2 derived lake heights against in situ gauge measurements from Australian Bureau of Meteorology (BoM) and Environment Canada (EC).

We validated two NRT approaches (i.e., (1) geostatistical model + extent (GSWD (Landsat) + geostatistical model + BLUEDOT (Sentinel-2)) and (2) geostatistical model + height (GSWD (Landsat) + geostatistical model + *V-H* relationship + ICESat-2)) to derive water storage time series from 1984 to present against in situ data that can help to assess the accuracy of both our historical and NRT methods (Fig. 10). We compared the performance of water storage estimates between the respective methods used to derive historical and NRT estimates. The average *R* and SMAPE for the method, evaluated for 1984-2020 with storage estimated using a geostatistical model, were 0.94 and 22%. The performance for the NRT methods, evaluated for 2020-present with storage estimated using a *V-H* relationship or *V-V* merging, decreased slightly to *R*=0.88 and SMAPE=31%, respectively. For most lakes, the agreement between predicted and observed storages shows similar accuracy (in terms of correlation and bias, Table 2) for the historical and NRT methods, as NRT estimates were only produced if the *V-H* or *V-A* relationships were significantly correlated. For example, both products record sharply decreasing in water storage in Big Sandy Reservoir, Vega Reservoir, and El Vado Lake in the NRT method period (Fig. 10). However, the performance of the NRT methods decrease for Lake Summer, Horsetooth Reservoir and El Vado Lake as there was one erroneous observation (Fig. 10).

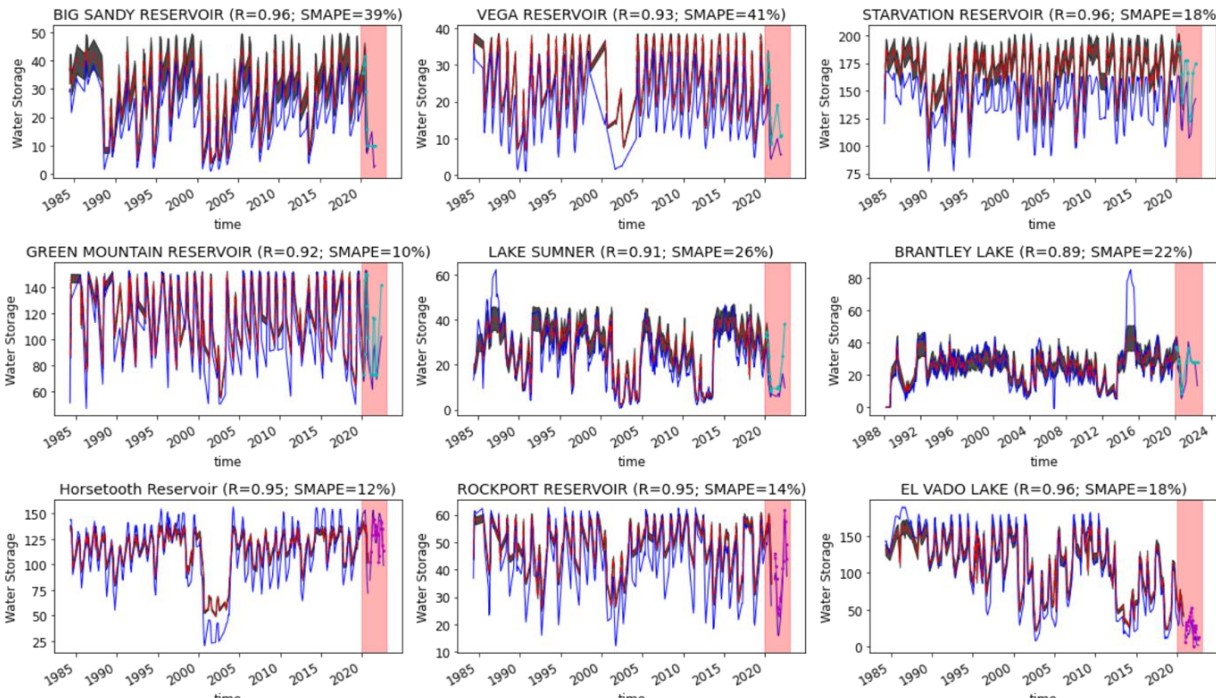

**Figure 10** Comparisons of historical and NRT lake water storage time series from GloLakes against in situ (observed) data (blue line: observed data; red line: historical storage estimates; black shade: error bars of historical storage estimates; light pink shade: NRT monitoring period; cyan line with dots (first two rows): NRT storage estimates from *V-H* relationship with ICESat-2; magenta line with dots (third row): NRT storage estimates from geo-statistical model with BLUEDOT (Sentinel-2)).

We compared NRT estimates from our different storage products and summarised the results for lakes of different sizes (Table 3). The mean *R* and SMAPE between GSWD (Landsat)/ICESat-2 and GSWD (Landsat)/BLUEDOT(Sentinel-2) products were 0.83 and 7%, respectively. The GSWD (Landsat)/ICESat-2 and GSWD (Landsat)/GREALM products had a similar correlation (*R*=0.81) and bias (SMAPE=7%), although they are both derived from altimetry. The evaluation results generally did not vary with lake size, except that GSWD (Landsat)/ICESat-2 and GSWD (Landsat)/BLUEDOT (Sentinel-2) products show better correlation (*R*=0.92) and less bias (SMAPE=2%) for lakes with an area greater than 100 km$^2$.

**Table 2** Comparisons of correlations and biases between historical and NRT methods (results from Fig. 10).

| Lake Name | NRT Monitoring Approach | Historical (1984-2020) | | NRT (2020-present) | |
|---|---|---|---|---|---|
| | | R | SMAPE | R | SMAPE |
| Big Sandy | V-H; ICESat-2 | 0.96 | 39% | 0.94 | 53% |
| Vega | | 0.93 | 41% | 0.98 | 52% |
| Starvation | | 0.96 | 18% | 0.97 | 18% |
| Green Mountain | | 0.92 | 10% | 0.93 | 13% |
| Sumner | | 0.91 | 26% | 0.81 | 43% |
| Brantley | geo-statistical model; BLUEDOT (Sentinel-2) | 0.89 | 22% | 0.89 | 21% |
| Horsetooth | | 0.95 | 12% | 0.56 | 8% |
| Rockport | | 0.95 | 14% | 0.99 | 19% |
| El Vado | | 0.96 | 18% | 0.81 | 48% |

**Table 3** Cross-validations of NRT methods between our different storage products for lakes of different sizes.

| | GSWD (Landsat) / ICESat-2 | | | | | | | |
|---|---|---|---|---|---|---|---|---|
| | GSWD (Landsat) / BLUEDOT (Sentinel-2) | | | | GSWD (Landsat) / GREALM | | | |
| | <10 km$^2$ | 10-100 km$^2$ | >100 km$^2$ | All | <10 km$^2$ | 10-100 km$^2$ | >100 km$^2$ | All |
| Total number of overlapped lakes | 1247 | 612 | 92 | 1951 | 7 | 22 | 63 | 92 |
| R | 0.83 | 0.83 | 0.92 | 0.83 | 0.84 | 0.84 | 0.80 | 0.81 |
| SMAPE | 8% | 7% | 2% | 7% | 7% | 11% | 5% | 7% |


## 3.5 Current limitations and future opportunities

Remote sensing provides an opportunity to measure water in most lakes worldwide and provide NRT information, which is impossible with the current in situ network. This study used the Landsat-derived GSWD to estimate lake surface water extents. The main limitation of using GSWD is that it only produced monthly observations from 1984-2020 and cannot provide NRT

information, despite its use of Landsat observations with a temporal frequency of 16 days. The GSWD are expected to be updated annually by the Joint Research Centre of the European Commission (https://global-surface-water.appspot.com/download), at which point lake extent estimates can be extended beyond 2020. In this study, we harnessed the capabilities of BLUEDOT (Sentinel-2) to expand lake surface water extent estimates beyond 2020, benefitting from the 5-day revisit interval of Sentinel-2 within the BLUEDOT framework. However, lake area data from BLUEDOT are presently

available only for a subset of several thousand lakes across the globe. This issue could be overcome using MODIS, VIIRS or Sentinel-2 data directly, for example. The daily or 8-day composite MODIS and VIIRS products have a better chance to

provide valid observations, but the hundred-meter range resolution is often not sufficient to accurately detect lake area changes. 5-day, 10-m resolution water extent derived from Sentinel-2 should be a promising candidate for NRT global lake monitoring. The sheer volume of data presented us with a challenge for data storage and processing, but there is no fundamental limitation
that would prevent a similar approach from measuring all 170,957 lakes measured by GSWD (Landsat) in this study. The advantage of Sentinel-2 would be that it can provide NRT lake observations with low latency. The high computation and storage demands could potentially be met by cloud platforms like Google Earth Engine (GEE). In future research, we hope to consider such approaches to improve our data set.

Topex/Poseidon (1992-2002), Jason 1/2/3 (2002-present), and Sentinel-6 (a.k.a. Jason-CS; 2020-present) are all able to measure lake height every ten days, which should be adequate to monitor dynamics in large lakes. Sentinel-3A/B (2016-present) has a revisit time of 27 days, still providing valuable, quasi- monthly updates on changes on water level. The dynamic estimates of lake height from these radar altimeters were seamlessly processed and derived from G-REALM. However, radar altimeters cannot detect many smaller lakes in between the sparse ground tracks. The ICESat-2 laser altimeter covers many
more lakes globally, benefiting from its dense reference tracks enhanced by the six laser beams onboard. The trade-off is its temporal resolution of ~91 days, but this is still sufficient to observe seasonal changes in many lakes worldwide, and more frequent water extent mapping can be used to interpolate between these observations.

Comparing altimetry and optical sensors to measure lake changes, the greatest challenge arises from optical remote sensing.
This is primarily due to the inherent limitations associated with optical remote sensing, notably the influence of clouds, various atmospheric interferences, and, to a lesser extent, vegetation. This issue could be mitigated by using passive microwave sensors or SARs. For example, the Japanese Space Agency's AMSR2 and TRMM TMI sensors and NASA's AMSR-E and GPM instruments can provide daily observations of surface water based on differences in brightness temperature between wet and dry areas (De Groeve et al., 2015; Hou et al., 2018). Unfortunately, their resolution is generally very coarse due to the
observation method. Sentinel-1 SAR could be a more practical solution to monitor lakes under cloud cover, with 12 days and 10-m resolution, provided the water detection algorithm can be automated, and vegetation cover does not interfere with the mapping. Finally, the Surface Water and Ocean Topography (SWOT) satellite mission was launched at the end of 2022 and is expected to measure surface water height and extent simultaneously every 11 days for lakes greater than 250 m by 250 m. Its temporal resolution is intermediate to the radar and laser altimetry used here, but SWOT shows promise for monitoring lake
changes given that the two basic components (i.e., $A$ and $H$) needed to estimate lake volume change are measured simultaneously.

## 4. Data availability

The GloLakes data described here are available from https://dx.doi.org/10.25914/K8ZF-6G46 (Hou et al., 2022). Six products are provided, described in Table 4. The products also provide additional attributes such as data quality (Q1: absolute volume estimated using geostatistical model and satellite-derived lake extents; Q2: absolute volume estimated based on the *V-H* relationship; Q3: relative volume estimated based on both satellite-derived heights and extents; Q4: relative volume estimated based on heights obtained from satellite measurements, combined with the extents of the lake derived from the area-height (*A-H*) relationship), latitude, longitude, lake name, country name, state/province name, basin name and catchment name for each lake. The products can be linked to the HydroLAKES database using the ID index provided in the metadata. This allows users to combine storage dynamics data with other lake attributes, such as lake type, shoreline length, hydraulic residence times, and watershed area, among others. Using the same HydroLAKES metadata, the data can also be coupled to the GRanD (Lehner et al., 2011), HydroSHEDS (Lehner et al., 2008), and HydroRIVERS (Lehner and Grill, 2013) databases. This allows users to relate other river and reservoir attributes and implement the data in hydrological model configuration, e.g., to improve river routing. The global lake dynamics can also be interactive explored through the Global Water Monitor (http://www.globalwater.online), which along with NRT information on other water cycle data such as river discharge (Hou et al., 2020; Hou et al., 2018), soil moisture (https://cds.climate.copernicus.eu/cdsapp#!/dataset/10.24381/cds.d7782f18?tab=overview, v202012 combined product) and precipitation and other meteorological variables (Beck et al., 2022).

**Table 4** Overview of GloLakes Product Descriptions

| Filename | Type of volume | Satellite sources | Historical method | NRT method | The number of measured lakes | Time Coverage |
|---|---|---|---|---|---|---|
| Global_Lake_Absolute_Storage_LandsatPlusGREALM (1984-present).nc | Absolute | GSWD (Landsat) + G-REALM | Q1 | Q2 | 129 | 1984-current |
| Global_Lake_Absolute_Storage_LandsatPlusICESat2 (1984-present).nc | Absolute | GSWD (Landsat) + ICESat-2 | Q1 | Q2 | 24,865 | 1984-current |
| Global_Lake_Absolute_Storage_LandsatPlusSentinel2 (1984-present).nc | Absolute | GSWD (Landsat) + BLUEDOT (Sentinel-2) | Q1 | Q1 | 4,054 | 1984-current |
| Global_Lake_Relative_Storage_LandsatPlusGREALM (1993-present).nc | Relative | GSWD (Landsat) + G-REALM | Q3 | Q4 | 227 | 1993-current |
| Global_Lake_Relative_Storage_LandsatPlusICESat2 (2018-present).nc | Relative | GSWD (Landsat) + ICESat-2 | Q3 | Q4 | 24,990 | 2018-current |

| Global_Lake_Relative_Storage_Sentinel2PlusICESat2 (2018-present).nc | Relative | BLUEDOT (Sentinel-2) + ICESat-2 | Q3 | Q3 | 2740 | 2018-current |
|---|---|---|---|---|---|---|

## 5. Conclusions

We produced historical and near real-time lake storage dynamics from 1984-present by combining optical remote sensing (i.e.,
Landsat and Sentinel-2) and radar and laser altimetry data (i.e., Topex/Poseidon, Jason-1/2/3, Sentinel-3/6, and ICESat-2) at
the global scale. The historical lake storage time series can help improve understanding of the influence of climate change and
human activities on global or regional lake storage dynamics from 1984 to the present. The NRT lake storage data we provide
will hopefully provide useful and current information for those managing our water resources and aquatic ecosystems. Surface
water extent time series were estimated for all HydroLAKES-delineated lakes larger than 1 km$^2$. To maximise measurement
frequency, a simple image gap-filling algorithm was implemented in each historical GSWD (Landsat) surface water map. This
process effectively restored missing data caused by, e.g., cloud, cloud shadow, swath edges and the Landsat-7 SLC failure.
The monthly gap-filling GSWD (Landsat) derived water extents showed strong correlations with Zhao and Gao (2018) for
5,318 reservoirs and Donchyts et al. (2022) for 11,101 lakes, with a median *R* of 0.91 and 0.76, respectively. The geostatistical
HydroLAKES bathymetry estimation approach produced slightly better volume estimates than the GLOBathy method and was
applied to estimate absolute lake volume dynamics from 1984-2020 for 170,611 lakes. Validation results showed a median *R*
of 0.91 and a SMAPE of 38% between estimated and reported volumes for 494 lakes in the USA, South Africa, India, Spain
and Australia. In situ bathymetric measurements would be needed to more accurately estimate total lake volume, but they are
not available for the majority of the millions of lakes globally. Relative lake volume storage changes were measured for lakes
where their satellite-derived extents and heights were both available. The median correlation between our relative storage
product and previous MODIS-derived data was 0.94. In addition, we investigated where and for how many lakes whose NRT
storage time series can be estimated by remote sensing around the world. The results indicated that the most suitable regions
for satellite-based lake monitoring include the USA, southeastern South America, the Mediterranean, southern Africa, southern
Asia, and Australia. NRT monitoring was achieved using BLUEDOT (Sentinel-2), ICESat-2 and GREALM. The validation
results showed that the performance of NRT storage estimation decreases slightly compared to historical estimates but are still
of good quality. In the future, the estimates could be improved using Sentinel-2-derived global surface water mapping and
observations from the recently launched SWOT mission.

**Author contribution.** JH and AIJMVD conceived the idea. JH developed the methodology and software, carried out the
investigation and analysis, curated the data and wrote the first manuscript draft. AIJMVD, LJR, and PRL reviewed and edited
the manuscript.

**Competing interests**. The contact author has declared that none of the authors has any competing interests.

**Acknowledgments.** The authors acknowledged the European Commission's Joint Research Centre, the National Snow & Ice

Data Center (NSIDC), US Department of Agriculture (USDA) and BLUEDOT for providing satellite products. We thanked

Australian Bureau of Meteorology, Environment Canada, US Bureau of Reclamation and US Army Corps of Engineers for

sharing in situ lake and reservoir data. Calculations were performed on the high-performance computing system, Gadi, from

the National Computational Infrastructure (NCI), which is supported by the Australian Government

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
