# Peer review of "GloLakes: water storage dynamics for 27,000 lakes globally from 1984 to present derived from satellite altimetry and optical imaging"

_Earth System Science Data, 2022_

## Author Comment (AC1)

**Response to Reviewer #1 Comments:**

Hou et al present a dataset of global water storage variations. This paper is unique in that it attempts to construct absolute storage variability time series, which are challenging to produce. It also aims to fuse together multiple freely available datasets in a novel approach. However, I find it to overall be a flawed manuscript and dataset which needs many necessary improvements (see major comments below). In brief, I am concerned that the authors have not accurately described the dataset, both in regards to the long term and NRT storage dynamics and have not performed sufficient validation analyses. The paper is also poorly written in places, with numerous typos and grammatical errors as well as paragraphs that are poorly structured, and the figures are weak and do not sufficiently illustrate the dataset. Without substantial changes to the manuscript, presentation and perhaps the dataset itself, I'm not sure this paper and dataset would be of value to the broader community.

We thank the reviewer for the detailed and valuable comments and suggestions, which will enable us to greatly improve the quality of our manuscript. Below please find our response to reviewer's comments in detail.

In the revised manuscript, we will carefully and accurately distinguish between "estimate" and "measure" to describe the different approaches to derive lake water storage time series (in response to comment R1C1). We will also add the results of several more validations analyses we have undertaken to strengthen the manuscript:

- (1) We extended our water storage validation analysis from 238 lakes to 494 lakes (now including in situ data from USA, Australia, South Africa, India, and Spain), and show correlation and bias results for all 494 lakes.
- (2) We performed uncertainty analysis of the geo-statistical model in the water storage validation analysis for 21 lakes as examples. Uncertainties sources include observed lake area (the omission and commission errors of surface water mapping from GSWD used in this study are 5% and 1%, respectively; Pekel et al. (2016)) and DEM (±10 m; Robinson et al. (2014)).
- (3) We compared lake mean volume from our product against in situ data for 494 lakes to examine any systematic bias in our data.
- (4) We validated near real-time (NRT) storage estimates for 9 lakes where in situ data include NRT information and analysed the change in the performance of storage estimation between NRT and historical estimates.
- (5) We cross-validated NRT storage estimates between our different approaches and show the performance of storage estimation for lakes with different sizes.
- (6) We compared lake water area time series from our product against Zhao and Gao (2018) and Donchyts et al. (2022) for 5318 and 11101 lakes, respectively, in terms of correlation and bias, and showed the comparisons of lake mean area between our product and these two published studies in the 1:1 relationship to examine the uncertainties of our derived lake area time series.
- (7) We provided a summary as to where and for how many lakes NRT storage can be estimated by remote sensing in each basin around the world. This will provide key information for the newly

launched Surface Water and Ocean Topography (SWOT) to monitor storage changes in global lakes as it will be able to measure both extent and level on a single satellite platform.

(8) We show examples of comparisons between our relative storage product and Tortini et al. (2020) and correlation results for all evaluated lakes.

We apologise for any typos and grammatical errors and will carefully examine and correct them. We will also add several more figures (please see the details in the response to the comments below) to better illustrate our product. Finally, we will restructure our Results and Discussion Section to have the following sections:

- (1) Lake area estimation validation
- (2) Where NRT lake monitoring is feasible
- (3) Validation of historical lake storage estimates
- (4) Validation of NRT lake storage estimates
- (5) Future opportunities

Following the review, we will also make some changes to our dataset in the revised manuscript:

- (1) We will implement the ICESat-2 quick look data into our GloLakes product to improve monitoring abilities of ICESat-2 based storage estimations
- (2) We will release a new product to derive lake storage changes directly using satellite-derived extents (Sentinel-2) and levels (ICESat-2)
- (3) The GREALM has now included Sentinel-6A data. This will be included our product as well

**Major Comments**

R1C1) Estimation of long term storage dynamics. Throughout the text, the authors state that they 'measured' long term absolute storage dynamics and/or 'produced' absolute storage time series. To be clear, what the authors did was apply a geostatistical model to estimate water depth and then use this to estimate a absolute storage time series when combined with a Landsat-derived dataset of lake extent. There was thus no 'measurement of storage' here – what the authors did was 'estimate' storage based on statistical relationships. While there is nothing wrong with estimating using these geostatistical relationships, it is imperative that this is explained correctly and consistently throughout the paper so as not to cause confusion with other methods which actually calculate volume change based on water level observations.

Agreed. We will carefully revise the manuscript and make sure that we consistently use "estimate" where a geostatistical model was used while retaining "measure" where storage is calculated from water extent and level observations, to indicate the lesser uncertainty in the latter.

R1C2) ICESat-2 data is not NRT. The authors state the importance of Near Real Time (NRT) lake monitoring with a latency of ~1-10 days. They then include ICESat-2 as one of the potential datasets to use for NRT monitoring. However, this indicates a fundamental misunderstanding of ICESat-2 and how it is processed. First of all, ICESat-2 has a repeat time of 91 days, so on average you get an observation ~once every three months

(though this does vary based on the size of the lake). Second of all, unlike with say MODIS, Landsat, or Sentinel-2, ICESat-2 data is not immediately released. Currently (as of Oct 9, 2022), the most recently available ICESat-2 data is through June 8th, 2022, and this has been fairly consistent over the past few years (ICESat-2 releases data about every ~6 months). While the NSIDC ICESat-2 website is perhaps a little misleading that it says data is available up to the present, so I understand some of the confusion, simple playing around with the data will quickly reveal the extremely long latency of ICESat-2 products. It is thus very much inaccurate to use ICESat-2 as a potential NRT water volume estimator in the method described here.

We agree that ICESat-2 cannot provide sufficient near real-time (NRT) information if we used the "ATLAS/ICESat-2 L3A Along Track Inland Surface Water Data, Version 5 (ATL13)" product because their data were not updated regularly when new observations obtained. In using the term NRT, we referred to the time between the ICESat-2 data becoming available and our use of it to produce storage estimates, but we appreciate this can be a misleading use of words. We also thank the reviewer for referring us to the "ATLAS/ICESat-2 L3A Along Track Inland Surface Water Data Quick Look, Version 5 (ATL13QL)" product which can provide NRT water height observations (R1C21). We will implement the ICESat-2 quick look data into our GloLakes product to improve monitoring abilities of ICESat-2 based storage estimations and make changes in the revised manuscript accordingly. We will include how we process ATL13 and ATL13QL data and combine them together to derive NRT water height observations and discuss the advantages and disadvantages of using these two products in the revised manuscript.

We were aware that ICESat-2 has a low temporal frequency of around 91 days. However, this kind of temporal frequency can still provide updated and useful seasonal variations of global lakes. In revising, we will clarify that NRT monitoring for ICESat-2 based storage estimations has latency to update each of lake storage but that we produce a rapid update global lake data *once* there are new observations.

Another important aspect of using ICESat-2 in this study was to investigate where lake storages can be measured by satellite-derived extent and level simultaneously, as it has very dense coverage of global lakes, thus overlapping a large number of lakes with Landsat observations. These will provide key information for the newly launched Surface Water and Ocean Topography (SWOT) to monitor storage changes in global lakes as it achieves to measure both extent and level on a single satellite platform. Please see detailed response to R1C5.

Based on all above, we will modify the paragraph on ICESat-2 data description in the revised manuscript:

"ATLAS/ICESat-2 L3A along-track inland surface water data, version 5 (ATL13) was used in this study (Jasinski and Ondrusek, 2021). The updated ATL13 product is not released until 30-45 days after new observations were obtained, which does not fit the purpose for the NRT monitoring. Therefore, in addition, we used ATLAS/ICESat-2 L3A Along Track Inland Surface Water Data Quick Look, Version 5 (ATL13QL), which is available within 3 days of new observations. ATL13 and ATL13QL apply the same algorithms to derive surface water height measurements for inland water bodies including rivers, lakes, reservoirs and coastal water. The approach is described by Jasinski and Ondrusek (2021), but in brief: 1) identifies ATLAS beams that intersect inland water body shape masks; 2) collects photons in short segments (~100 m) for calculating water height and in longer segments (1~3 km) for estimating and removing subsurface backscatter; 3) applies the physical and statistical modelling to derive inland water heights. Cooley et al. (2021) used ATLAS/ICESat-2 L3A Land and Vegetation Height (ATL08) to derive lake water height time series and found the mean absolute error between USGS gauge data and ICESat-2 height measurements is 0.14 m. Unlike ATL08, ATL13 used in this study was designed to measure water height variations on rivers and lakes as it considers physical processes of light propagation in open water bodies. An error of 6.1 cm per 100 inland water photons has been reported (Jasinski and Ondrusek, 2021). We collected both ATL13 and ATL13QL water surface height data from any of the six beams within individual lake boundaries from HydroLAKES and derived water height time series for each lake that observed by ICESat-2 from 2018 to present. ATL13QL has larger uncertainties (~100 m) in geolocation than ATL13 (~5 m). As a result, segment heights from ATL13QL are 2.7 m, with a standard deviation of  $\sim$  7 m, lower than those from ATL13. According to the user guide (Jasinski and Ondrusek, 2021), 2.7 m can be added to ATL13QL before merging these two products and the differences between them have little impact on measuring relative heights. Each lake will be revisited by ICESat-2 around every 91 days. This means our ICESat-2 based storage product is limited to provide updated seasonal variations, rather than short-term changes, for global lakes. However, it provides updated data where there are new observations with a rapid turnaround."

R1C3) No validation of NRT data. While the authors do perform validation for 238 lakes for what appears to be the geostatistically estimated historical time series, it is not explicitly stated whether they perform any validation of the NRT time series (both the absolute and relative). More detailed information on the accuracy of the NRT time series, and how it varies between using V-H vs. V-A relationships (or how it varies by lake size, if possible), is required to be able to evaluate this dataset.

We thank the reviewer for this comment. It is a challenge to find publicly accessible in situ data especially with NRT information. Of course, this is one of the reasons we are using remote sensing in the first place. We would like to first emphasize that NRT storages were only estimated if V-H or V-A relationships were significantly correlated (please see the detailed response to R1C5). After a search, we found NRT data for nine sites and used these to validate our NRT product. In addition, we performed cross-validation between NRT estimates from our different storage products (ICESat-2, Sentinel-2, and GREALM), and summarized results for lakes with different sizes. We will include these new validation results and a new paragraph in the revised manuscript as follows:

"We validated our ICESat-2 derived and Sentinel-2 derived water storage time series from 1984 to present against in situ data that contains both historical and NRT observations (Fig. R1). We compared the performance of water storage estimates between historical and NRT periods to investigate if NRT estimation based on V-H or V-A relationships is valid. The average R and SMAPE in the historical period (1984-2020) if storage was estimated using a geostatistical model were 0.94 and 22%. The performance for the NRT period (2020-present) when storage was estimated using a V-H or V-A relationships decreased slightly (R=0.88 and SMAPE=31%). For most lakes the agreement between predicted and observed storages in NRT period shows similar accuracy (in terms of correlation and bias, Table R1) as NRT storages in our products were only estimated if V-H or V-A relationships were significantly correlated. For example, both ICESat-2 and Sentinel-2 derived products record sharply decreasing in water storage in Big Sandy Reservoir, Vega Reservoir, and El Vado Lake in the NRT period (Fig. R1). However, the performance for the NRT period statistically decreases for Lake Summer Horsetooth Reservoir and El Vado Lake as there is one erroneous observation (Fig. R1).

We compared NRT estimates from our different storage products (ICESat-2, Sentinel-2, and GREALM), and summarized results for lakes with different sizes (Table R2). The mean R and SMAPE between ICESat-2 and Sentinel-2 products was 0.77 and 12%, respectively. The ICESat-2 and GREALM products had a similar correlation (R=0.75) but less bias (SMAPE=6%) as they are both derived from altimetry. The evaluation results generally did not vary as a function of lake size except for the fact that ICESat-2 and Sentinel-2 products show better correlation (R=0.84) and less bias (SMAPE=5%) for lakes with an area greater than 100 km2."

---

## Author Comment (AC2)

**Response to Reviewer #2 Comments:**

Hou et al. presented a nice study on creating a new time-varying dataset on global lake water storage. Understanding the lake storage variablity is critical for securing freshwater supplies. The authors leveraged multiple datasets to produce a global dataset with hundreds of thousands of lakes included, which seems to be an impressive work. However, I have a few major comments on the method and data quality.

**We thank the reviewer for the thoughtful comments and constructive suggestions, which will help us to improve the quality of the manuscript. Below please find our response to reviewer's comments in detail.**

R2C1) Pekel et al. used a global model to classify water and land. The authors depended on the Pekel et al.'s data to track water area changes in each lake locally. It is not clear to me whether the global model is suitable for studying each individual water body, for example, how can lake area change be accurately captured by the global model in each lake?

The GSWD dataset developed by Pekel et al. (2016) has been widely used to derive surface water area dynamics for individual lake in many studies (for example, Zhao and Gao (2018); Busker et al. (2019)). These previous studies have demonstrated the validity of using the GSWD dataset to study lake and reservoir changes. Pekel et al. (2016) also validated their product. The results showed that the omission errors of water mapping are less than 5%, while the commission errors are less than 1%. In the revised manuscript, we will include these uncertainties from GSWD in the storage validation analysis. Please refer to the response to R2C3. In revision, we will also include comparisons between our lake area change estimation and other published lake area datasets (i.e., Zhao and Gao (2018) and Donchyts et al. (2022)). Please see response to R2C2.

Most importantly, this study focuses on estimating both historical and near real-time (NRT) lake water storage dynamics. NRT lake water storage time series were only estimated if A-H-V relationships were significantly correlated, although we produced historical water storage change for more than 170,000 lakes (Fig. R5). This also means we make sure that satellite-derived extent and level observations are consistent in characterising lake change before producing storage estimates. We will include a paragraph to highlight this point in the revised manuscript:

"This study mainly focuses on estimating both historical and NRT lake water storage dynamics. NRT lake water storage time series were only estimated if A-H-V relationships were significantly correlated, although we produced historical water storage change for more than 170,000 lakes. This also means we make sure that satellite-derived extent and level observations are consistent in characterising lake change before producing lake storage time series from 1984 to present.

As Landsat, Senitnel-2 and ICESat-2 used here have comprehensive coverage of global lakes, we investigated if we can use any two of them (optical + altimetry or two optical) to monitor global lakes and for how much of the lakes in each basin storage can be measured by remote sensing. This should provide valuable information for the newly launched Surface Water and Ocean Topography (SWOT) mission that monitors storage changes in global lakes as it achieves to measure both extent and level on a single satellite platform. If remote sensing measurements from two sources (derived extent and level or

derived storages) are significantly correlated for a particular lake, we considered that this lake can be monitored by Earth observation. We followed three complimentary approaches: (1) geo-statistical models (Landsat + Sentinel-2), (2) H-V relationships (Landsat + ICESat-2), (3) extent and level observations (Sentinel-2 + ICESat-2). Approach (3) can be considered the most reliable as storages were directly measured by extent and level from remote sensing. However, approaches (1) and (2) make it possible to estimate absolute storage changes. Landsat and ICESat-2 together are able to measure lake water storage in nearly all (i.e., 234) river basins worldwide (Fig. R5b). Satellite-derived extents and levels are significantly correlated for over one fourth of lakes in each of the 145 basins. This feature is evenly distributed across the continents except for Antarctica and northern high latitude regions, due to the influence of frozen water surfaces. Sentinel-2 and ICESat-2 cover 122 basins globally (Fig. R5d). There are 58 basins where over half of lakes can be monitored by them, mainly located in the USA, southeastern South America, the Mediterranean, southern Africa, southern Asia, and Australia. Landsat and Sentinel-2 both measure surface water extent and they show consistency in 63 out of 124 basins. Three-quarters of lakes has a significant H-A relationship (Fig. R5c) with a distribution pattern similar to that of Sentinel-2/ICESat-2."

**Figure R5** The locations of 170,957 lakes whose storage dynamics for the period of 1984-2020 were estimated in this study (a), the percentages of lakes (in terms of number) whose NRT water storage dynamics can be derived using Landsat and ICESat-2 (b), Landsat and Sentinel-2 (c), and Sentinel-2 and ICESat-2 (d) in each basin.

**[1] Donchyts, G., Winsemius, H., Baart, F., Dahm, R., Schellekens, J., Gorelick, N., Iceland, C., & Schmeier, S. (2022). High-resolution surface water dynamics in Earth's small and medium-sized reservoirs. Scientific reports, 12(1), 1-13.**

R2C2) The monthly lake areas were generated from recovering water areas from contaminated images as in a previous publication by the authors (Hou et al., 2022). This is not new as quite a few recent studies have done a similar thing. As monthly lake areas are critical for generating monthly storage given monthly level data is pretty rare, the uncertainty of the recovered water areas seems to have non-negligible impact on the derived storage change. Had the authors assessed the uncertainty of areas from contaminated images? How did the generated time series compare with other existing approaches?

We thank the reviewer for this suggestion. In revising the manuscript, we will include comparisons between our lake area change estimates and other published lake area datasets (i.e., Zhao and Gao (2018) and Donchyts et al. (2022)) (Fig. R7) as below:

"Our lake area data have 5318 lakes in common with the data of Zhao and Gao (2018), and 11101 lakes with Donchyts et al. (2022). Zhao and Gao (2018) used the same Landsat data source (GSWD) and the same lake boundary delineation (GRanD included in HydroLAKES) but a different gap-filling approach to derive lake area. Nonetheless, the mean lake area between the two products scatters closely around a 1:1 relationship (Fig. R7a). In addition, we calculated correlation and bias in lake area time series for the common period 1984–2018 for each lake. The median R and SMAPE were 0.91 and 3.6%, respectively. This suggests that our gap-filling algorithm produces results that are overall similar to those of Zhao and Gao (2018). Donchyts et al. (2022) used a different Landsat data source, lake boundary delineation, and gap-filling algorithm to derive lake area time series. Despite these differences, mean lake area values still lies fairly closely around the 1:11 relationship, especially for lakes greater than 1 km2 (Fig. R7b). Larger biases exist for some lakes smaller than 1 km2. This was caused mainly by different definitions of lake boundaries between HydroLAKES and Donchyts et al. (2022); we found only one-tenth of lakes had boundary area differences within 20%. The median R and SMAPE in lake area time series from 1984 to 2020 for 11101 lakes between our product and Donchyts et al. (2022) were 0.76 and 9.7%, once again lower mainly due to the differences for small lakes."

Figure R7 Scatterplots of mean lake area from our product vs. two other published datasets (black line: 1:1 relationship).

R2C3) I do not believe the Geo-statistical model used by Messager et al., 2016 to predict the total volume of a lake can be used to derive actual lake bathymetry here. The Geo-statistical model was based on global DEM products which have an uncertainty of several meters on average. Additionally, the water level conditions at the DEM acquisition time vary. As the used DEM data in Messager et al., 2016 cannot retrieve the true land surface elevation underneath water, I think this would introduce an even larger uncertainty (e.g., dozens of meters) when the authors extrapolated water levels beneath the level at the DEM acquisition date.

We agree that lake volume cannot be estimated very precisely unless detailed lake bathymetry data is available, but this is beyond current satellite remote sensing abilities. We emphasized this point in the manuscript. However, we respectfully disagree that the approach used in this study is affected by water level condition at DEM acquisition time, as the approach is different from the traditional DEM approach. The traditional DEM approach indeed is affected by water level condition at DEM acquisition time as it used DEM within the lake to derive A-H curves. In comparison, the model used here considers DEM outside lake (i.e., extrapolating slope around the lake towards the centre of the lake to estimate lake depth) rather than inside, therefore it is not affected by water inundation at the DEM acquisition time.

We did compare different approaches to estimate lake depth before choosing the geo-statistical model, but did not include the full analysis in this manuscript. Khazaei et al. (2022) develop a statistical model relating lake depth with surface water area, elevation, volume, shoreline length, and watershed area. They demonstrated that the performance of this statistical model is better than the previous approaches to calculate lake depth by simply assuming lake shape in four geometries: box, cone, triangular prism, and ellipsoid (e.g., Yigzaw et al. 2018). In our study, we compared this statistical model (Khazaei et al., 2022) with the geostatistical model (Messager et al., 2016) and compared them against in situ data. The resulting SMAPE error for the GLOBahty method was 70.5%, and therefore not better than the geostatistical method. This led our choice to use the geo-statistical method in favour of the GLOBathy dataset in this study.

[2] Yigzaw, W., Li, H. Y., Demissie, Y., Hejazi, M. I., Leung, L. R., Voisin, N., & Payn, R. (2018). A new global storage-area-depth data set for Modeling reservoirs in land surface and earth system models.
Water Resources Research, 54(12), 10-372.

The geo-statistical model used in this study has uncertainties sources from DEM and observed lake area as demonstrated in Equation R1. The GSWD data used in this study to derive lake area has the omission and commission errors of 5% and 1%, respectively. The DEM used in geo-statistical model to calculate slope has an error of around 10 m. We have considered all these uncertainties in our storage validation results. In the revised manuscript, we will include a new paragraph (please see below) to better describe the geo-statistical model and include uncertainties analysis (Fig. R2).

"The geo-statistical model (Messager et al., 2016) used in this study is:

 $Log_{10}(D) = C_1 + C_2 \times Log_{10}(A) + C_3 \times Log_{10}(S_{100}) + s^2$ (R1)

where D is the predicted mean depth (m), A the observed surface area of the lake (km2), S100 the average slope (derived from DEM) within a 100-m buffer around the lake, C1, C2 and C3 constant parameters estimated from best fitting the model using global lake data and present for different sizes of lakes (i.e., 0.1-1 km2, 1-10 km2, 10-100 km2, 100-500 km2), and s2 the residual variance. The fundamental assumption is that one can extrapolate slope around the lake towards the centre of the lake to estimate lake depth. The traditional approach uses a DEM within the lake to derive A-H curves, but cannot retrieve true land surface elevation below water. In comparison, the model used here considers the DEM outside lake rather than inside, and hence is not affected by water inundation at the time of DEM acquisition. Messager et al. (2016) reported that the symmetric mean absolute percent error between predicted and reference volume is 48.8% without significant bias in volumes for the majority of lakes around the world, with the exception of Finland, Sweden and northwestern Russia, the European Alps, and the Andes. The uncertainties from the geo-statistical model are mainly from observed surface area

and DEM used to derive slope. The omission and commission errors of surface water mapping from GSWD used in this study are 5% and 1%, respectively. EarthEnv-DEM90 was used in the geo-statistical model to derive slope, and its vertical accuracy is around 10 m (Robinson et al., 2014). We assessed how these errors propagate into water storage estimation in the validation analysis."

---

## Author Response (AR1)

**Response to Reviewers**

**''GloLakes: water storage dynamics for 27,000 lakes globally from 1984 to present derived from satellite altimetry and optical imaging'' by Jiawei Hou et al.**

We thank the two reviewers for their thoughtful comments and constructive suggestions, which helped us improve the manuscript significantly. We have thoroughly considered all comments and suggestions, and made modifications accordingly below (review comments in blue, our responses in black bold font).

In summary, we added several validation analyses in the revision below:

(1) We extended our water storage validation analysis from 238 lakes to 494 lakes (now including in situ data from USA, Australia, South Africa, India, and Spain), and showed correlation and bias results for all 494 lakes.

(2) We performed uncertainty analysis of the geo-statistical model in the water storage validation analysis for 21 lakes as examples. Uncertainties sources include observed lake area (the omission and commission errors of surface water mapping from GSWD used in this study are 5% and 1%, respectively; Pekel et al. (2016)) and DEM (±10 m; Robinson et al. (2014)).

(3) We compared lake mean volume from our product against in situ data for 494 lakes to examine any systematic bias in our data.

(4) We validated near real-time (NRT) storage estimates for 9 lakes where in situ data include NRT information and analysed the change in the performance of storage estimation between NRT and historical methods.

(5) We cross-validated NRT storage estimates between our different approaches and showed the performance of NRT storage estimation for lakes with different sizes.

(6) We compared lake water area time series from our product against Zhao and Gao (2018) and Donchyts et al. (2022) for 5318 and 11101 lakes, respectively, in terms of correlation and bias, and showed the comparisons of lake mean area between our product and these two published studies in the 1:1 relationship to examine the uncertainties of our derived lake area time series.

(7) We provided a summary as to where and for how many lakes NRT storage can be estimated by remote sensing in each basin around the world. This will provide key information for the newly launched Surface Water and Ocean Topography (SWOT) to monitor storage changes in global lakes as it will be able to measure both extent and level on a single satellite platform.

(8) We showed examples of comparisons between our relative storage product and Tortini et al. (2020) and correlation results for all evaluated lakes.

We rephrased several paragraphs in Abstract, Introduction, Data and method and Conclusion sections, and reconstructed Results and Discussion Section to have the following sections:

(1) Lake area estimation validation
(2) Feasibility of near real-time lake monitoring
(3) Validation of historical lake storage estimates
(4) Validation of near real-time lake storage estimates
(5) Future opportunities

**Following the reviews, we also made some changes to our dataset:**

(1) **We implemented the ICESat-2 quick look data into our GloLakes product to improve monitoring abilities of ICESat-2 based storage estimations**

(2) **We released a new product to derive lake storage changes directly using satellite-derived extents (Sentinel-2) and levels (ICESat-2)**

(3) **The GREALM has now included Sentinel-6 data. This was included in our product as well**

**Response to Reviewer #1 Comments:**

Hou et al present a dataset of global water storage variations. This paper is unique in that it attempts to construct absolute storage variability time series, which are challenging to produce. It also aims to fuse together multiple freely available datasets in a novel approach. However, I find it to overall be a flawed manuscript and dataset which needs many necessary improvements (see major comments below). In brief, I am concerned that the authors have not accurately described the dataset, both in regards to the long term and NRT storage dynamics and have not performed sufficient validation analyses. The paper is also poorly written in places, with numerous typos and grammatical errors as well as paragraphs that are poorly structured, and the figures are weak and do not sufficiently illustrate the dataset. Without substantial changes to the manuscript, presentation and perhaps the dataset itself, I'm not sure this paper and dataset would be of value to the broader community.

**We thank the reviewer for the detailed and valuable comments and suggestions, which enabled us to greatly improve the quality of our manuscript. Below please find our response to reviewer's comments in detail.**

**In the revised manuscript, we carefully and accurately distinguished between "estimate" and "measure" to describe the different approaches to derive lake water storage time series. We also added the results of several more validation analyses to strengthen the manuscript.**

**We apologise for any typos and grammatical errors. We carefully examined and corrected them in the revised manuscript. We also added several more figures and restructured paragraphs in several sections to better illustrate our product.**

**Major Comments**

R1C1) Estimation of long term storage dynamics. Throughout the text, the authors state that they 'measured' long term absolute storage dynamics and/or 'produced' absolute storage time series. To be clear, what the authors did was apply a geostatistical model to estimate water depth and then use this to estimate a absolute storage time series when combined with a Landsat-derived dataset of lake extent. There was thus no 'measurement of storage' here – what the authors did was 'estimate' storage based on statistical relationships. While there is nothing wrong with estimating using these geostatistical relationships, it is imperative that this is explained correctly and consistently throughout the paper so as not to cause confusion with other methods which actually calculate volume change based on water level observations.

**Agreed. We carefully revised the manuscript and make sure that we consistently use "estimate" where a geostatistical model was used while retaining "measure" where storage is calculated from water extent and level observations, to indicate the lesser uncertainty in the latter. Please see the changes in the revised manuscript.**

R1C2) ICESat-2 data is not NRT. The authors state the importance of Near Real Time (NRT) lake monitoring with a latency of ~1-10 days. They then include ICESat-2 as one of the potential datasets to use for NRT monitoring. However, this indicates a fundamental misunderstanding of ICESat-2 and how it is processed. First of all, ICESat-2 has a repeat time of 91 days, so on average you get an observation ~once every three months (though this does vary based on the size of the lake). Second of all, unlike with say MODIS, Landsat, or Sentinel-2, ICESat-2 data is not immediately released. Currently (as of Oct 9, 2022), the most recently available ICESat-2 data is through June 8th, 2022, and this has been fairly consistent over the past few years (ICESat-2 releases data about every ~6 months). While the NSIDC ICESat-2 website is perhaps a little misleading that it says data is available up to the present, so I understand some of the confusion, simple playing around with the data will quickly reveal the extremely long latency of ICESat-2 products. It is thus very much inaccurate to use ICESat-2 as a potential NRT water volume estimator in the method described here.

**We agree that ICESat-2 cannot provide sufficient near real-time (NRT) information if we used the "ATLAS/ICESat-2 L3A Along Track Inland Surface Water Data, Version 5 (ATL13)" product because their data were not updated regularly when new observations obtained. In using the term NRT, we referred to the time between the ICESat-2 data becoming available and our use of it to produce storage estimates, but we appreciate this can be a misleading use of words. We also thank the reviewer for referring us to the "ATLAS/ICESat-2 L3A Along Track Inland Surface Water Data Quick Look, Version 5 (ATL13QL)" product which can provide NRT water height observations (R1C21). In the revision, we implemented the ICESat-2 quick look data into our GloLakes product to improve monitoring abilities of ICESat-2 based storage estimations and made changes in the revised manuscript accordingly. We also included how we process ATL13 and ATL13QL data and combine them together to derive NRT water height observations and discussed the advantages and disadvantages of using these two products in the revised manuscript.**

**We were aware that ICESat-2 has a low temporal frequency of around 91 days. However, this kind of temporal frequency can still provide updated and useful seasonal variations of global lakes. In revising, we clarified that NRT monitoring for ICESat-2 based storage estimations has latency to update each of lake storage but that we produce a rapid update global lake data *once* there are new observations.**

**Another important aspect of using ICESat-2 in this study was to investigate where lake storages can be measured by satellite-derived extent and level simultaneously, as it has very dense coverage of global lakes, thus overlapping a large number of lakes with Landsat observations. These will provide key information for the newly launched Surface Water and Ocean Topography (SWOT) to monitor storage**

**changes in global lakes as it achieves to measure both extent and level on a single satellite platform. Please see detailed response to R1C5.**

**Based on all above, we modified the paragraph in L134-154 in Section 2.1.2 (Surface water height) in the revised manuscript:**

*"We used the ATLAS/ICESat-2 L3A Along-Track Inland Surface Water Data, Version 5 (ATL13) (Jasinski and Ondrusek, 2021). Unfortunately, updated ATL13 data are not released until 30-45 days after new observations are obtained, which does not suit NRT monitoring. Therefore, we also used ATLAS/ICESat-2 L3A Along Track Inland Surface Water Data Quick Look, Version 5 (ATL13QL), available within three days of new observations. ATL13 and ATL13QL apply the same algorithms to derive surface water height measurements for inland water bodies, including rivers, lakes, reservoirs and coastal water. The approach is described by Jasinski and Ondrusek (2021), but in brief, the procedure: 1) identifies ATLAS beams that intersect inland water body shape masks; 2) collects photons in short segments (~100 m) for calculating water height and in longer segments (1~3 km) for estimating and removing subsurface backscatter; 3) applies the physical and statistical modelling to derive inland water heights. Cooley et al. (2021) used ATLAS/ICESat-2 L3A Land and Vegetation Height (ATL08) to derive lake height time series and found the mean absolute error between USGS gauge data and ICESat-2 height measurements is 0.14 m. Unlike ATL08, the ATL13 used here was designed to measure water height variations in rivers and lakes as it considers physical processes of light propagation in open water bodies. An error of 6.1 cm per 100 inland water photons has been reported (Jasinski and Ondrusek, 2021). We collected both ATL13 and ATL13QL water surface height data from any of the six beams within individual lake boundaries from HydroLAKES and derived water height for each lake observed by ICESat-2 between 2018 and the present. ATL13QL has larger uncertainties (~100 m) in geolocation than ATL13 (~5 m). As a result, segment heights from ATL13QL are 2.7 m, with a standard deviation of ~ 7 m, lower than those from ATL13. According to the user guide (Jasinski and Ondrusek, 2021), 2.7 m can be added to ATL13QL before merging these two products, and the differences between them have little impact on measuring relative heights. Each lake will be revisited by ICESat-2 around every 91 days. This means our ICESat-2-based storage product is limited to providing information on seasonal variation rather than short-term dynamics. However, it has the benefit of providing observations with a rapid turnaround."*

R1C3) No validation of NRT data. While the authors do perform validation for 238 lakes for what appears to be the geostatistically estimated historical time series, it is not explicitly stated whether they perform any validation of the NRT time series (both the absolute and relative). More detailed information on the accuracy of the NRT time series, and how it varies between using V-H vs. V-A relationships (or how it varies by lake size, if possible), is required to be able to evaluate this dataset.

**We thank the reviewer for this comment. It is a challenge to find publicly accessible in situ data especially with NRT information. Of course, this is one of the reasons we are using remote sensing in the first place.**

We would like to first emphasize that NRT storages were only estimated if V-H or V-A relationships were significantly correlated (please see the detailed response to R1C5). After a search, we found NRT data for nine sites and used these to validate our NRT product. In addition, we performed cross-validation between NRT estimates from our different storage products (ICESat-2, Sentinel-2, and GREALM), and summarized results for lakes with different sizes (please note this result is a bit different from the response in the discussion phase as there are updates and changes in the datasets). We included these new validation results and added a new section 3.4 (Validation of near real-time lake storage estimates) in the revised manuscript as follows:

[revised manuscript text omitted]

R1C4) Need for large error bars and more details about the approach. The paper glosses over the specifics of the statistical method (pointing towards another paper) used to contruct the historical time series, but more details on this method should be provided to enable the reader to better understand the method and its potential accuracy. The dataset should also include significant error bars on the resulting time series, or at least clear information that these are all estimates with on average ~50% error. This is even more important for the NRT estimations, as given the use of look-up tables to approximate V-H/V-A relationships, these should have even greater error than the original volume time series (see comment about need for validation above)!

**Thank you for these suggestions. We included error bars considering the uncertainties from the geo-statistical model (Fig. 5 and 9). And we modified Section 2.1.3 (Geostatistical model) to better explain the method and how we assess its uncertainties:**

*"The HydroLAKES database provides the boundary outlines of more than 1.4 million individual lakes globally with a surface area above 0.1 km² (Messager et al., 2016). This database is compiled from several global and regional lake and reservoir datasets derived using topographic maps, optical remote sensing imagery composites and radar instruments. HydroLAKES also comprises a geostatistical model with parameters to predict average water depths and volumes based on surrounding topography information for lakes with a surface extent between 0.1 km² and 500 km². The geostatistical model (Messager et al., 2016) is described as follows:*

[revised manuscript text omitted]

R1C5) Poor quality figures. The figures included are weak and do not provide sufficient info about the dataset. Figure 3 is very difficult to interpret (what is the difference between panel (b) and panel (c))? Figures 1 and 2 are useful but are very large and all 4 examples for each may not be needed. Figure 4 is useful and depicts the accuracy analysis well. I'd suggest adding additional figures illustrating the accuracy of the geostatistical estimation of water depth (and perhaps the resulting volume time series). It also would be useful to provide

more information about where is was possible to build NRT and relative time series, and where it was not possible (perhaps in a figure? Or table?).

**We apologise for the confusion. Fig. 3b and 3c showed the percentage of total basin water volume for which water storage dynamics were estimated in this study. The percentage is calculated in each basin by Equation R1:**

$$p = \frac{\sum_{i=0}^{m} V_i}{\sum_{i=0}^{n} V_i} \quad (R1)$$

**where *p* is the percentage of total lake water volume in each basin, *m* the number of lakes for which water storage dynamics were estimated in a basin, *n* the total number of lakes from HydroLAKES in a basin, and $V_i$ the water volume of individual lake from HydroLAKES. The difference between these two figures is that Fig. 3b counted the lakes that have been observed by either optical remote sensing or altimetry instruments while Fig. 3c calculated those whose historical time series were estimated.**

**In the revised manuscript, we deleted Fig. 3b and 3c and added Fig. 3b-d to emphasize one of highlights in this study. To make it easier to interpreted, we calculated the percentage of total number of lakes rather than total volume explained above in the new Fig.R3b-d. Fig. R3b-d shows where and for how much of the lakes NRT storages can be measured by remote sensing in each basin around the world. These will provide valuable information for the newly launched Surface Water and Ocean Topography (SWOT) to monitor storage changes in global lakes as it achieves to measure both extent and level on a single satellite platform. We added a new section 3.2 (Feasibility of near real-time lake monitoring) to explain this analysis in the revised manuscript:**

*"We estimated monthly lake volume dynamics for 170,611 lakes during 1984–2020 (Fig. 3a). Most lakes are in the northern hemisphere's high latitudes, especially in North America. We only considered lakes with an area greater than 1 km² given the resolution of the Landsat imagery. There are more than a million lakes with areas between 0.1–1 km² that remain unmeasured, but the combined storage of these small lakes only accounts for 1% of global total lake water storage, according to the HydroLAKES dataset. Overall, a large portion of global lake volume for the period 1984-2020 was measured. Here we focus on estimating both historical and NRT lake dynamics. NRT lake water storage was only estimated if A-H-V relationships were sufficiently strong, even though we produced historical water storage change for more than 170,000 lakes. This also ensures that satellite-derived extent and level observations are consistent in characterising lake change before producing lake storage time series from 1984 to the present.*

*As the Landsat, Sentinel-2 and ICESat-2 have comprehensive coverage of global lakes, we investigated if any two of them (optical + altimetry or two optical) can be used to monitor global lakes and how much of the lakes in each basin storage can be measured in each case. This should provide valuable information for the newly launched Surface Water and Ocean Topography (SWOT) mission that aims to monitor storage changes in global lakes by measuring both extent and level on a single satellite platform. The*

*HydroBASINS database delineates watershed boundaries worldwide at different basin or catchments scales (Lehner and Grill, 2013). The catchment boundaries in the Pfafstetter level-3 product from HydroBASINS were used as the spatial basin units for analysis. If remote sensing measurements from two sources (derived extent and level or derived storages) are significantly correlated for a particular lake, we considered that this lake can be monitored by Earth observation. We followed three complementary approaches: (1) geostatistical model (Landsat + Sentinel-2), (2) H-V relationships (Landsat + ICESat-2), and (3) extent and level observations (Sentinel-2 + ICESat-2). Approach (3) would likely be the most reliable as storage is directly measured by extent and level from remote sensing. However, approaches (1) and (2) make it possible to estimate absolute storage changes. Landsat and ICESat-2 together could measure lake water storage in nearly all (i.e., 234 out of 292) river basins worldwide (Fig. 3b). Satellite-derived extents and levels are significantly correlated for more than one-fourth of lakes in each of the 145 basins. This feature is evenly distributed across the continents, except for Antarctica and high northern latitudes, due to the influence of frozen water surfaces. Sentinel-2 and ICESat-2 cover 122 basins globally (Fig. 3d). There are 58 basins where over half of the lakes can be monitored by that method, mainly located in the USA, southeastern South America, the Mediterranean, southern Africa, southern Asia, and Australia. Landsat and Sentinel-2 both measure surface water extent and show strong time series consistency in 63 out of 124 basins. Three-quarters of lakes have a significant A-H relationship (Fig. 3c) with a distribution pattern similar to that of Sentinel-2/ICESat-2."*

[Figure]

**Figure 3** The locations of 170,611 lakes whose storage dynamics for the period of 1984-2020 were estimated in this study (a), the percentages of lakes (in terms of number) whose NRT water storage dynamics can be derived using Landsat and ICESat-2 (b), Landsat and Sentinel-2 (c), and Sentinel-2 and ICESat-2 (d) in each basin.

**On the other hand, Fig.1 and Fig.2 showed how our gap-filling algorithm addressed missing Landsat data in different scenarios (cloud cover, SLC failure, mix issues, different contamination ratios, different shapes of lake). Following the reviewer's comment, we clarified this in L280-283 in the revised manuscript and moved Fig. 2 to the Supplemental Material.**

**Besides we showed examples (Fig. 8) of comparisons between our relative storage product and Tortini et al. (2020) and correlation results (Fig. 7) for all evaluated lakes in L382-387 in the revised manuscript as follows:**

> *"We compared our relative lake storage dynamics data against the global surface water storage change time series for 1992–2018 produced by Tortini et al. (2020) for 104 common lakes. The data was downloaded from the Physical Oceanography Distributed Active Archive Center (PODAAC) of NASA's Jet Propulsion Laboratory. The two show strong agreement (Fig. 7) with a median R of 0.94 and 95% of R values above 0.7. Selected time series are compared in Fig. 8 and show that both datasets are most limited by the temporal frequency of image acquisition."*

[Figure]

**Figure 7** The distribution of R of relative water storage validation results against Tortini et al. (2022) for 104 lakes (vertical blue line: the median value).

[Figure]

**Figure 8** Relative lake water storage time series from GloLakes (blue line) against Tortini et al. (2022) (brown line) for selected lakes.

R1C6) Why not include Landsat and Sentinel-2 (not BLUEDOT) as NRT? Given that the authors calculate NRT storage variations simply by building V-H or V-A relationships with the results of their Landsat and

geostatistical depth model volume time series, it should be possible to construct NRT time series for thousands more lakes globally by just classifying water in Landsat-8/9 and Sentinel-2 (and this would actually be NRT, unlike ICESat-2) While I understand that part of the point of this paper is to use existing datasets, classifying water in Landsat-8/9 and Sentinel-2 is pretty darn standard and straightforward at this point, particularly with the existence of cloud platforms like GEE. I'm not suggesting the authors do this globally, more just pointing out that this approach (while likely quite inaccurate) could in theory be used for thousands more lakes. Also, why (for the Sentinel-2 data in particular) do you need to use a lookup table to estimate volume via a V-A relationship – surely you could use the same geostatistical model to calculate volume from the Sentinel-2 (or Landsat 8/9) area observation?

**Thank you for this suggestion. Regards to GEE, we agree that we could directly use Sentinel-2 to derive water extent for thousands more lakes, but it was indeed well beyond the scope of this study. In the revised manuscript, we discussed the limitation of using BLUEDOT data and how this might be addressed using Sentinel-2 and GEE in future in L452-456 in the revised manuscript as follows:**

> *"Sentinel-2-based lake area data are currently already available from BLUEDOT for several thousand lakes worldwide, and there is no fundamental limitation that would prevent a similar approach from measuring all 170,957 lakes measured by Landsat in this study. The advantage of Sentinel-2 would be that it can provide NRT lake observations with low latency. The high computation and storage demands can potentially be met by cloud platforms like Google Earth Engine (GEE). In future research, we hope to consider such approaches to improve our data set."*

**We would also like to clarify that we did in fact use the geostatistical model to estimate water storage for both of Landsat and Sentinel-2 and then merged them together to derive historical and NRT water storage time series. We modified Section 2.2.2 (Global near real-time lake volume estimation) to clarify this point and all the other NRT storage estimation approach in the revised manuscript as follows:**

> *"To provide routine and low-latency measurements of lake water storage, we obtained NRT satellite-derived water heights and extents from different monitoring satellite sources, including ICESat-2), US Department of Agriculture (USDA) GREALM and BLUEDOT (Sentinel-2) water observatory (Table 1). We selected all estimates for lakes whose volume dynamics from 1984–2020 were derived from GSWD with the geostatistical model. We examined pairwise relationships (i.e., V-H or V-A) of overlapping monthly time series between historical volume (V) and NRT height (H) or area (A). We only extended historical volume estimation to NRT monitoring when there was a statistically significant (p<0.05) correlation between the predictor and predictand. Specifically, we calculated the significant Pearson correlation threshold ($R_t$) using a t-test allowing for sample size N (i.e., the number of data pairs). If the Pearson correlation (R) between V-H or V-A exceeded $R_t$ we used H or A, respectively, to estimate lake storage dynamics after 2020.*

*Depending on some of the features of the satellite data sources (Sentinel-2, ICESat-2, GREALM), we used different approaches to estimate absolute NRT storage. If both historical and NRT sources (i.e., Landsat and Sentinel-2) only observed lake area changes, we converted lake area to storage using the geostatistical model (Eq. 1) and merged them to derive storage time series from 1984-present. If NRT satellite sources (i.e., ICESat-2 or GREALM) observed lake water height changes, we developed the relationship between historical volume (V) and NRT height (H) for the overlapping months and used that V-H to extend historical volume estimation to NRT monitoring using only satellite-derived H. As the relationship is not necessarily linear, NRT lake storage was estimated using cumulative distribution function (CDF) matching. A look-up table was developed to rank all historical H and V, allowing one to be estimated from the other based on the ranking. Overall, we derived absolute water storage dynamics from 1984 onwards for 24,865 lakes using ICESat-2, 129 lakes using GREALM, and 4,054 lakes using Sentinel-2.*

*Our main objective was to estimate NRT absolute water storage dynamics for lakes worldwide as much as possible. However, we also measured relative storage where radar or laser-derived surface water heights, as well as Landsat-derived surface water extent, were available (i.e., Landsat with ICESat-2 or Landsat with GREALM). Unlike absolute storage products, relative storage products are unaffected by any bias from the geostatistical model. We produced alternative storage datasets to provide options to users requiring different bias tolerances or different applications. For the relative storage products, we calculated historical storage changes for lakes where the A-H relationship was significantly correlated as follows (Crétaux et al., 2016):*

$$\Delta V = \frac{(H_t - H_{t-1}) \times (A_t + A_{t-1} + \sqrt{A_t \times A_{t-1}})}{3} \quad (2)$$

*where $\Delta V$ is storage change between two consecutive measurements; $H_t$ and $H_{t-1}$ are altimetry-derived surface water heights at time t and t-1, respectively; $A_t$ and $A_{t-1}$ are optical remote sensing derived surface water extents at time t and t-1, respectively. Where there are only H data available in the NRT period, we estimated corresponding A using CDF matching and calculated relative storage change time series using Eq (2). As Sentinel-2 and ICESat-2 can measure NRT surface water extent and level, respectively, we used them to derive relative storage changes from 2018-present. This product is free from any errors in the geostatistical model and in the A-H or V-H relationships used in the other products. Overall, we measured relative lake storage dynamics from 2018 onwards for 24,990 lakes using ICESat-2 and Landsat, for 2,740 lakes using ICESat-2 and Sentinel-2, and from 1993 onwards for 227 lakes using GREALM and Landsat."*

**Considering that Sentinel-2 and ICESat-2 can measure surface water extent and level, respectively, simultaneously, we produced an additional data set as part of this collection to derive lake storage changes directly rather than using the A-H or V-H relationships. Please see the details above. Below are some examples (Fig. R1) of H-A (Sentinel-2 + ICESat-2) relationships for the overlapping period (2016-present).**

[Figure]

**Figure R1** Some examples of lake *H-A* (Sentinel-2 and ICESat-2) relationships from 2016-present (blue line: fitting relationships; blue shade: 95% confidence interval; top axis: the distribution of water level; right axis: the distribution of water area).

**Specific Comments**

R1C7) Lines 6-7: The first sentence of the abstract is unnecessarily long and wordy and contain a typo. Please rephrase.

**Thank you. We shortened this sentence in L6-7 in the revised manuscript:**

> *"Measuring the spatiotemporal dynamics of lake and reservoir water storage is fundamental for assessing the influence of climate variability and anthropogenic activities on water quantity and quality."*

R1C8) Lines 56-74: Nice review of all of the different global water datasets, but this paragraph could use some restructuring as in its current form, this transition from the statement about ICESat-2 laser altimetry towards stating that "the spatial resolution of global satellite-dervied surface water dynamics projects…" doesn't make much sense, as the paragraph then moves to talking about measurements of surface water extent, not water level. I would suggest reorganizing this paragraph and being clearer about developments in observations of water level vs. water extent.

**Apologies for the confusion. These are two separate paragraphs and we modified them in the revised manuscript.**

R1C9) Line 85: Add "cannot measure lake depth and therefore are unable to measure absolute water volume without the use of bathymetric data"

**Thank you. We added this in L86 in the revised manuscript.**

R1C10) Lines 93-105: This paragraph is confusingly worded. The authors state that "Relative storage changes were estimated … while absolute storage changes were…" which is immediately followed by a statement that this was possible for more than 27,000 lakes worldwide. According to my understanding of the paper, the absolutely storage changes estimated using a geostatistical model were possible for all ~170,000 lakes, whereas the relative changes were possible for ~23,000 and NRT volumes (using the statistical model plus V-H or V-A relationships) were estimated for ~27,000. As currently written, this paragraph thus inaccurately describes the results.

**Thank you. We rephrased this paragraph in L95-112 in the revised manuscript:**

*"The objectives of this study were to (1) derive nearly four decades of data on relative and absolute water volume measurements for lakes worldwide and (2) enable a global NRT lake monitoring capability. To develop this, we first estimated water body extents between 1984-2020 from Landsat-derived surface water maps for 170,957 lakes. We applied the gap-filling algorithm (Hou et al., 2022) in contaminated Landsat images to restore missing data, thus improving the total number of usable images to derive lake area time series. Second, we estimated absolute water storage dynamics from 1984-2020 for each lake whose water area and its surrounding slope measurements are available using a geostatistical model (Messager et al., 2016). Third, as Landsat does not provide NRT observations, we considered a range of alternative satellite data sources (including Sentinel-2, Jason-3, Sentinel-3 and -6, and ICESat-2) with monitoring abilities to derive NRT absolute lake storage. We examined where and for how many lakes NRT storage can be estimated using different combinations of these remote sensing data in each basin worldwide. Fourth, we extended historical absolute water storage estimates to NRT monitoring using the volume-height relationship if radar or lidar altimetry data available after 2020. Where only NRT lake water area observations (e.g., from Sentinel-2) were available, we converted lake area to storage estimates using a geostatistical model. As these absolute lake storage products are affected by the bias errors of lake depth estimates from the geostatistical model, we also provided relative lake storage estimates for lakes observed simultaneously by optical imagers and altimetry. Overall, this made it possible to monitor more than 27,000 lakes and reservoirs worldwide. The underlying strategy in developing this global lake monitoring system was to consider all readily available, validated, and frequently updated satellite data sources, explore the relative advantage of each source, and combine them to complement their respective weaknesses."*

R1C11) Line 97: Remove "Furthermore"

**Thank you. We removed "Furthermore" as this sentence has been changed in the revised manuscript.**

R1C12) Line 153: What is meant by "future GSWD water bodies"?

**Apologies for the confusion. We can simply remove "future" in L195 in the revised manuscript**

R1C13) Line 195: While NSIDC is where the ICESat-2 data is hosted, it is incorrect to call it a monitoring platform. Just call it ICESat-2 data.

**Thank you. We called it ICESat-2 data consistently in the revised manuscript**

R1C14) Table 1: It is incorrect to call NSIDC the platform for ICESat-2. I'm not sure exactly what term would be useful here, but I'd suggest simply calling the datasets something like ICESat-2, USDA G-REALM and BLUEDOT (Sentinel-2).

**Agreed. We used ICESat-2, USDA G-REALM and BLUEDOT (Sentinel-2) as their names in the revised manuscript.**

R1C15) Figure 3 caption (and throughout the manuscript). It is incorrect to state that "storage dynamics for the period of 1984-2020 were measured in this study". Given that storage was estimated based purely on statistical relationships that are likely to be highly inaccurate in many places (and with an overall error of ~50%), 'estimated' is the only appropriate word to describe this approach.

**Thank you. We carefully corrected and used either "estimate" or "measure" to describe different approaches (see response to R1C1).**

R1C16) Line 277: What is meant by "Overall, the relative volume dynamics are generally more reliable, as indicated by the correlation values"? Does this refer to the NRT time series? Or specifically the NRT time series calculated from A-H relationships? And correlation with what? The results on this accuracy are not reported in the paper (see major comment above).

**Apologies. By this we mean that the validation results (predicted volume vs. observed volume) showed strong correlation so our product should be more reliable to indicate lake volume changes than absolute depth changes. We also produced relative storage products that avoid using the geo-statistical model and focus on providing accurate relative changes in lake storage. In addition, we added validations of the NRT estimates in the revised manuscript (see response to R1C3).**

R1C17) Line 291: Again, please be clear here that these results are 'estimated'.

**Agreed, see R1C15.**

R1C18) Line 299: It is not necessary to state that Busker's dataset was not publicly available (you can always just email an author and ask for it, as technically every dataset should be publicly available on some level). I suggest removing this part of the sentence.

**Agreed. We removed this in the revised manuscript.**

R1C19) Line 343: SWOT will measure water height every 11 days (its orbit has a return period of 21 days, but since it is an off-nadir satellite, it will measure every water body every 11 days – see https://swot.jpl.nasa.gov/

**Thank you. We changed it to "11 days" in the revised manuscript.**

R1C20) Line 356: I explored the online Global Water Monitor linked in the paper. This is a cool way of displaying and communicating the data, and I commend the authors for putting it together, though it needs some cleaning up a bit (for example, what is the unit "GL" on the y axis for the annual time series of water volume)? If possible, it also would benefit from including error bars.

**Thank you for this suggestion. The first version of the Global Water Monitor (GWM) has been released in January 2023 and we have updated the access link to web-based data explorer (www.globalwater.online) in the revised manuscript. Beyond this manuscript, we will consider all comments and suggestions and improve GWM in the next version in future.**

**Additional Comments:**
R1C21) I wanted to lightly addend my review, specifically about the latency of ICESat-2, as I just learned about the existence of ICESat-2 Quick Look, low-latency products (https://nsidc.org/data/user-resources/help-center/faqs-icesat-2-quick-looks). My comments about the limited value of ICESat-2 for NRT data due to its 91 day repeat cycle still apply, but the existence of these quick look products does mean that it is technically possible to use the data within a few days after collection. I apologize for not acknowledging this before.
If the authors are to continue to use ICESat-2 data for their NRT dataset, I would advise explicitly discussing the quick look data as well as its advantages/disadvantages relative to the final product. Similarly, while the other two datasets mentioned in the NRT section (G-REALM and BLUEDOT) are relatively easy to download and process into lake height/area (since they already come processed to individual lake height/area) doing so with ICESat-2 is significantly more complicated as it requires choices around how to aggregate different tracks, filter out poor quality or outlier water level observations, etc. This additional difficulty should be described in the manuscript with additional details on how the authors process the ICESat-2 data automatically.

**Thank you for these addition comments, see response to R1C2.**

**Response to Reviewer #2 Comments:**

Hou et al. presented a nice study on creating a new time-varying dataset on global lake water storage. Understanding the lake storage variablity is critical for securing freshwater supplies. The authors leveraged multiple datasets to produce a global dataset with hundreds of thousands of lakes included, which seems to be an impressive work. However, I have a few major comments on the method and data quality.

**We thank the reviewer for the thoughtful comments and constructive suggestions, which helped us to improve the quality of the manuscript. Below please find our response to reviewer's comments in detail.**

R2C1) Pekel et al. used a global model to classify water and land. The authors depended on the Pekel et al.'s data to track water area changes in each lake locally. It is not clear to me whether the global model is suitable for studying each individual water body, for example, how can lake area change be accurately captured by the global model in each lake?

**The GSWD dataset developed by Pekel et al. (2016) has been widely used to derive surface water area dynamics for individual lake in many studies (for example, Zhao and Gao (2018); Busker et al. (2019)). These previous studies have demonstrated the validity of using the GSWD dataset to study lake and reservoir changes. Pekel et al. (2016) also validated their product. The results showed that the omission errors of water mapping are less than 5%, while the commission errors are less than 1%. In the revised manuscript, we included these uncertainties from GSWD in the storage validation analysis. Please refer to the response to R1C4. In revision, we also included comparisons between our lake area change estimation and other published lake area datasets (i.e., Zhao and Gao (2018) and Donchyts et al. (2022)). Please see response to R2C2.**

**Most importantly, this study focuses on estimating both historical and near real-time (NRT) lake water storage dynamics. NRT lake water storage time series were only estimated if *A-H-V* relationships were significantly correlated, although we produced historical water storage change for more than 170,000 lakes (Fig. R5). This also means we make sure that satellite-derived extent and level observations are consistent in characterising lake change before producing storage estimates. We added a new section 3.2 (Feasibility of near real-time lake monitoring) to highlight this point in the revised manuscript. Please also see the detailed changes in R1C5.**

R2C2) The monthly lake areas were generated from recovering water areas from contaminated images as in a previous publication by the authors (Hou et al., 2022). This is not new as quite a few recent studies have done a similar thing. As monthly lake areas are critical for generating monthly storage given monthly level data is pretty rare, the uncertainty of the recovered water areas seems to have non-negligible impact on the derived storage change. Had the authors assessed the uncertainty of areas from contaminated images? How did the generated time series compare with other existing approaches?

**We thank the reviewer for this suggestion. In the revised manuscript, we included comparisons between our lake area change estimates and other published lake area datasets (i.e., Zhao and Gao (2018) and Donchyts et al. (2022)) (Fig. R7) in section 3.1 (Lake area estimation validation) as below:**

*"Our lake area data have 5318 lakes in common with the data of Zhao and Gao (2018) and 11,101 lakes with Donchyts et al. (2022) with average areas from 0.1 km² to 1000 km². Zhao and Gao (2018) used the same Landsat data source (GSWD) and the same lake boundary delineation (GRanD included in HydroLAKES) but a different gap-filling approach to derive lake area. Nonetheless, the mean lake area between the two products scatters closely around a 1:1 relationship (Fig. 2a). We also calculated correlation and bias in lake area time series for the common period 1984–2018 for each lake. The median R and SMAPE were 0.91 and 3.6%, respectively. This suggests that our gap-filling algorithm produces results overall similar to those of Zhao and Gao (2018). Donchyts et al. (2022) used a different Landsat data source, lake boundary delineation, and gap-filling algorithm to derive lake area time series. Despite these differences, mean lake area values still cluster fairly closely around the 1:1 relationship, especially for lakes greater than 1 km² (Fig. 2b). Larger biases exist for some lakes smaller than 1 km². This was caused mainly by different definitions of lake boundaries between HydroLAKES and Donchyts et al. (2022); we found that only one-tenth of lakes had boundary area differences within 20%. The median R and SMAPE in lake area time series from 1984 to 2020 for 11,101 lakes between our product and Donchyts et al. (2022) were 0.76 and 9.7%, once again lower mainly due to the differences for small lakes."*

[Figure]

**Figure 2** Scatterplots of mean lake area from our product vs. two other published datasets (black line: 1:1 relationship).

R2C3) I do not believe the Geo-statistical model used by Messager et al., 2016 to predict the total volume of a lake can be used to derive actual lake bathymetry here. The Geo-statistical model was based on global DEM products which have an uncertainty of several meters on average. Additionally, the water level conditions at the DEM acquisition time vary. As the used DEM data in Messager et al., 2016 cannot retrieve the true land surface elevation underneath water, I think this would introduce an even larger uncertainty (e.g., dozens of meters) when the authors extrapolated water levels beneath the level at the DEM acquisition date.

**We agree that lake volume cannot be estimated very precisely unless detailed lake bathymetry data is available, but this is beyond current satellite remote sensing abilities. We emphasized this point in L378-379 in the manuscript. However, we respectfully disagree that the approach used in this study is affected by water level condition at DEM acquisition time, as the approach is different from the traditional DEM approach. The traditional DEM approach indeed is affected by water level condition at DEM acquisition**

time as it used DEM within the lake to derive A-H curves. In comparison, the model used here considers DEM outside lake (i.e., extrapolating slope around the lake towards the centre of the lake to estimate lake depth) rather than inside, therefore it is not affected by water inundation at the DEM acquisition time. We highlighted this point in Section 2.1.3 (Geostatistical model) in the revised manuscript.

We did compare different approaches to estimate lake depth before choosing the geo-statistical model, but did not include the full analysis in this manuscript. Khazaei et al. (2022) develop a statistical model relating lake depth with surface water area, elevation, volume, shoreline length, and watershed area. They demonstrated that the performance of this statistical model is better than the previous approaches to calculate lake depth by simply assuming lake shape in four geometries: box, cone, triangular prism, and ellipsoid (e.g., Yigzaw et al. 2018). In our study, we compared this statistical model (Khazaei et al., 2022) with the geostatistical model (Messager et al., 2016) and compared them against in situ data. The resulting SMAPE error for the GLOBahty method was 70.5%, and therefore not better than the geo-statistical method. This led our choice to use the geo-statistical method in favour of the GLOBathy dataset in this study. We emphasised this analysis in the Section 2.1.3 (Geo-statistical model) in the revised manuscript.

[3] Yigzaw, W., Li, H. Y., Demissie, Y., Hejazi, M. I., Leung, L. R., Voisin, N., & Payn, R. (2018). A new global storage-area-depth data set for Modeling reservoirs in land surface and earth system models. Water Resources Research, 54(12), 10-372.

The geo-statistical model used in this study has uncertainties sources from DEM and observed lake area as demonstrated in Equation 1. The GSWD data used in this study to derive lake area has the omission and commission errors of 5% and 1%, respectively. The DEM used in geo-statistical model to calculate slope has an error of around 10 m. We have considered all these uncertainties in our storage validation results. In the revised manuscript, we modified Section 2.1.3 (Geostatistical model) to better describe the geo-statistical model and included uncertainties analysis in Fig. 5 and 9. Please also see the detailed changes in R1C4.

R2C4) The validation appears to be insufficient. How did the authors select the 238 lakes for the validation and why this is a comprehensive evaluation? Did the authors consider the performance of the method on cold-region lakes (e.g., in Canada). What the accuracy for smaller lakes given this method is only significant on small lakes as existing studies did a fairly good estimate on large lakes. Why the authors use relative metric R given R only gives a correlation estimate? For example, if the storage was scaled from area, no matter how large the level error is, the R value remains the same. Without a comprehensive validation, it is hard to foresee that the produced datasets would be useful for scientific inquiries.

Thank you for this comment. In the revised manuscript, we added several more validations analyses. Please see the details in the revision summary at the beginning of the author's response.

Unfortunately, we could not find any in situ data in Canada or other cold regions, although we have extended our storage validation results to cover more regions around the world. However, we assessed where and how much of lakes in each basin whose storage can be measured by remote sensing. The

results showed that only a modest percentage of lakes can be monitored by our approach in northern high latitude regions due to the influence of frozen water surfaces in the cold season (see response to R1C5).

We assessed the accuracy of our products for lakes of a wide range of sizes. The evaluations not only include validation of lake area estimates (see response to R2C2), but also lake storage (see Fig. 6 in R1C4). The lakes for which area estimates were validated had average areas from 0.1 km$^2$ to 1000 km$^2$ while those for which storage was validated had volumes from 1 GL to 10000 GL. We emphasised these in L293 and L358-359 in the lake area and storage validation sections in the revised manuscript.

We calculated not only R (correlation) but also SMAPE (bias) to assess the accuracy of our product. Storage was not just scaled from area, in fact, monitoring approaches include three types: (1) geo-statistical model (e.g., Landsat + Sentinel-2); (2) H-V relationships (e.g., Landsat + ICESat-2); (3) direct extent and level (e.g., Sentinel-2 + ICESat-2). We clarified this in L328-331 in the revised manuscript.

Specific comments:

R2C5) Line 64: I would suggest replacing "a larger number" with the actual number.

Thank you. The actual number was 227,386. We changed this in the revised manuscript.

R2C6) Line 86: It seems that Avisse et al. is not the only study on estimating lake storage change based on DEM data. Maybe also highlight other relevant studies here.

Thank you. We included the additional studies below in the revised manuscript.

[4] Bonnema, M., Sikder, S., Miao, Y., Chen, X., Hossain, F., Ara Pervin, I., Mahbubur Rahman S.M., & Lee, H. (2016). Understanding satellite-based monthly-to-seasonal reservoir outflow estimation as a function of hydrologic controls. Water Resources Research, 52(5), 4095-4115.

[5] Vu, D. T., Dang, T. D., Galelli, S., & Hossain, F. (2022). Satellite observations reveal 13 years of reservoir filling strategies, operating rules, and hydrological alterations in the Upper Mekong River basin. Hydrology and Earth System Sciences, 26(9), 2345-2364.

R2C7) Line 280: the authors only show a few case studies for the validation. It would be better to have a figure or table to show all data used for the validation.

Thank you, agreed. We showed R and SMAPE validation results for all 494 lakes (Fig. 4; see response to R1C2) in the revised manuscript.

---

## Author Response (AR3)

**Response to Reviewers**

**''GloLakes: water storage dynamics for 27,000 lakes globally from 1984 to present derived from satellite altimetry and optical imaging'' by Jiawei Hou et al.**

**We thank the two reviewers for their invaluable time and dedicated effort in reviewing our manuscript. We have carefully considered all comments and suggestions provided by the reviewers and have made modifications accordingly (review comments in blue, our response in black bold font). In summary, the major changes include:**

(1) **We carefully addressed all terminology issues that might lead to misinterpretation throughout the manuscript.**

(2) **We added quality labels in the GloLakes dataset to clarify the NRT methods. Data users can interpret from the meta-data how the data were generated. For example, one can readily choose the gap-filled GSWD-geostatistical model dataset by simply opting Q1 label or a time range spanning from 1984 to 2020.**

(3) **We enhanced the discussion on the revisit time of different datasets used in this study.**

(4) **In order to enhance the validation analysis of our lake extent estimates, we used Google Earth Engine to manually delineate lake extents based on Sentinel-2 data and compared to our lake extent estimates derived from GSWD (Landsat).**

(5) **We collected in situ lake level data from Australian Bureau of Meteorology and Environment Canada and used them to validate altimetry data used in this study. We paid more attention to the cold region in both lake extent and height validation analysis in the revised manuscript.**

**Response to Reviewer #1 Comments:**

The authors have responded to all reviewer comments in detail and clearly have taken care to improve the manuscript, and I commend them for their hard work and attention to detail. The manuscript has become clearer in places, including the description of the datasets and the greater emphasis on validation. However, the manuscript has a few areas where it still needs significant improvement.

First, there are important terminology issues that need to be addressed. As the manuscript is currently written, it is extremely misleading, as it does things like claim that Landsat is not NRT but ICESat-2 is, and also claim that ICESat-2 observes way more lakes than Sentinel-2. These things are true in the context of the datasets that are used in this paper (i.e. GSWD for Landsat, BLUEDOT for Sentinel-2, QL data for ICESat-2) but are not true generally, and so this language needs to be updated throughout the paper to prevent the paper from reading like a fundamental misunderstanding of remote sensing data. There's more detail in the comments below, but I'd suggest referencing the specific dataset, NOT the sensor, throughout the paper to clear up some of this confusion. Similarly, the authors refer to the GSWD-geostatistical model volume time series as the 'Landsat' time series in multiple places (including in Figure 3), which is also highly misleading and confusing. I also am still unsure about the value of the NRT analysis in this paper, and feel it may weaken the paper compared to the

GSWD-geostatistical model dataset. Lastly, the paper is missing some much needed discussion/results around frequency of observation and revisit times.

**Thank you so much for pointing out the weakness of this manuscript. We tried our best to mitigate these issues in the revised manuscript and please see our detailed responses below.**

Major Comments

R1C1) 1. Problematic use of data language and descriptions
There are significant issues with how the datasets are described throughout the paper. I understand that the reason so few lakes can be measured with Sentinel-2 is because the authors are not actually performing any classification themselves, but rather using the BLUEDOT dataset. I am not asking the authors to perform this classification, but I do think that every time the authors use the term 'Sentinel-2' in this paper, they should replace it with 'BLUEDOT'. It just seems very misleading to claim that ~24,000 lakes can be estimated in NRT with ICESat-2 vs. only ~4,000 lakes with Sentinel-2.

For one, this is really comparing apples to oranges – ICESat-2 has a revisit time of ~91 days, whereas Sentinel-2's is every 5 days, so time series estimated from Sentinel-2 would be much denser and more valuable. Not acknowledging this distinction throughout the paper is disingenuous. The 'science' value of a with a 91-day NRT revisit vs. a 5-day NRT revisit is wildly different, so this should be explicitly stated.

And secondly, it isn't the actual Sentinel-2 dataset that is being analyzed, rather a derived product. If the authors had actually classified all Sentinel-2 imagery over the HydroLAKES dataset, they would be able to build NRT time series for all ~170,000 lakes for which they estimated volume in the historical time series. Therefore, the results described here are quite misleading as they way understate the value of Sentinel-2 and way overstate the utility of ICESat-2 for NRT monitoring. At the very least, along with changing the terminology, these two issues (i.e. the difference in revisit time and the fact that the Sentinel-2 actually observes all lakes) should be discussed thoroughly in the discussion section.

**Apologies for the potentially misleading wording. In the revised manuscript, we replaced "Sentinel-2" with "BLUEDOT (Sentinel-2)" to clarify both data sources. We also clarified the limitations of BLUEDOT data used in this study and the abilities of Sentinel-2 to observe all lakes around the world in L517-523 in the revised manuscript. In addition, we modified Section 3.5 about the differences in revisit time of different datasets used in this study and their original remote sensing observations as well (Please see the detailed response in R1C4).**

**We agree that we could classify all Sentinel-2 imagery and combine it with GSWD (Landsat) to produce both historical and NRT volume dynamics for all ~170,000 lakes. Thank you for this suggestion. We believe this could be a future study and implemented in a next version of GloLakes. We have highlighted this in L522-528 in the revised manuscript.**

R1C2) Additionally, as noted in a specific comment below, the authors refer to the practice of deriving H-V relationships for the NRT data as 'Landsat-ICESat-2', but this is not a good way to describe what they are doing. They are (1) building a volume time series by combining the gap-filled JRC-GSWO (Landsat) dataset with the geostatistical model and then (2) correlating this with ICESat-2 observations to produce volume estimates from each ICESat-2 observation. This needs to be explicitly stated, and calling it a V-H correlation a 'Landsat-ICESat-2' correlation is highly misleading, because *Landsat does not measure volume*! This is also extra confusing when the authors describe V-A correlations as 'Landsat-Sentinel-2' since technically those both measure area…

**Thank you for pointing this out. In the revised manuscript, we clarified that V represents the GSWD or BLUEDOT-geostatistical model volume time series and avoided mentioning GSWD (Landsat) and BLUEDOT (Sentinel-2) when describing V-H-A relationships or water volume.**

R1C3) 2. Concerns about the NRT approach
I still feel very conflicted about the NRT method. If I'm understanding it correctly, the first part of the NRT method works by assessing the correlation between the historical volume data (which itself is based on a geostatistical model) and either NRT height or area, and then simply using a lookup table to convert the NRT data into volume time series. Most studies trying to use area and height data to calculate volume are specifically building curves between area and height and then extrapolating those (which is also done in this paper), whereas the geostatistical lookup table approach (i.e. correlated area OR height with estimated volume which itself is based on the combination of a geostatistical model and area observations), while applicable to large numbers of lakes, just seems to be overly complicated, especially as it is currently explained.

Also, I'm not sure what the value of this NRT analysis is (especially the ICESat-2 part), as it feels like it would be easier to just classify lake extent from any number of NRT optical sensors and then relate these to the geostatistical model to calculate volume if *actual* NRT data were needed, rather than trying to Frankenstein all these other datasets together, especially in way that significantly overstates the value of ICESat-2 and understates the value of Sentinel-2. If the goal is just to produce a global dataset of consistent estimates of lake volume, why not just wait until the GSWD data are updated? What value does having an NRT observation every ~91 days actually provide? (especially when you could just classify Sentinel-2 data and then have an observation for EVERY lake every 5 days…).

Overall, I understand and appreciate the authors' approach here – take a bunch of publicly available datasets, try putting them all together – but the end result as currently presented is misleading in places and I do question its scientific value (i.e. what will the NRT data actually be useful for). I personally think the paper and dataset would be stronger if the NRT section were removed entirely and this paper/dataset focused more on the fusion of the gap-filled GSWD data with the geostatistical model (which is a valuable contribution to the community), but I will of course leave that to the authors' and editor's discretion.

Thank you for your comments and suggestions. The main purpose of this study is to develop full time series of water storage estimates from 1984-present and to cover as many global lakes as possible by combining different freely accessible and regularly updated satellite data sources. We believe that the GloLakes dataset provides valuable long-term and updated information to understand past and current influences of climate change and variability and water management on natural lakes and reservoirs. We understand the reviewer's concern about the use of the term NRT for ~91 days old ICESat-2 observations and therefore replaced "NRT" with "up-to-date" for the ICESat-2 relevant estimates. We do argue that up-to-date (semi-seasonal, i.e. ~91 days) lake information from ICESat-2 is still valuable, however. We added a paragraph to highlight the value of the NRT (or up-to-date) lake information in L341-347 in the revised manuscript:

> "The primary objective behind generating the NRT lake data is to evaluate present-day conditions in lakes and dams in their historical context. This becomes takes on ever greater importance in an era of rapid climate change. Of particular significance is the storage volume, as it enables the aggregation of numerous water bodies within a single system. Such an aggregation offers a more comprehensive and insightful perspective on the hydrological status of catchments or river systems, such as provided by initiatives like the Global Water Monitor (www.globalwater.online). Knowledge of combined stored volumes are also needed to understand concepts such as catchment water equilibrium and to interpret GRACE terrestrial water storage estimates for example, enhancing our understanding of global water availability."

We appreciate your comment that the gap-filled GSWD-geostatistical model volume time series can be a valuable contribution to the community. We concur that our objective (namely, the estimation of NRT volume time series encompassing a significant portion of global lakes) can be accomplished through the classification of Landsat or Sentinel-2 imagery to delineate lake extents, followed by the application of a geo-statistical model to convert these areas into volumes. We hope to address this in our future research and include it in the next version of the GloLakes dataset.

We apologise for the confusion around the NRT methods, especially using the "lookup table". We argue that cumulative distribution function (CDF) matching as used in this study serves a similar purpose as building curves for V-A-H relationships. However, CDF matching is simpler compared to the traditional approach that involves selecting and fitting empirical equations (such as linear or nth degree polynomial equations). We modified the sentences in L256-263 to clarify CDF matching approach:

> "Unlike the conventional approach to build V-A-H curves, which typically requires the selection and fitting of empirical equations (e.g., linear or higher-degree polynomial equations), CDF matching takes a simpler route. It entails the development of a monotonic cumulative distribution function for historical height (H) and volume (V) data. The distribution allows for the ranking of values, showing their relative positions in the dataset. When a new height value (H) is to be retrieved, CDF matching uses the pre-

*established cumulative distribution functions and their associated rankings. By comparing the rank of the new value with those in the distributions, the corresponding volume value (V) can be determined."*

**We decided not to entirely remove NRT estimates, as data users already have the flexibility to focus on the 1984-2020 timeframe in which period lake volume estimates were exclusively generated through the gap-filled GSWD (Landsat) and geostatistical model (Table 4). In addition, there is one product whose both historical and NRT lake volume dynamics were entirely estimated based on the geo-statistical model (Table 4). However, to clarify the different NRT methods and their corresponding uncertainties, we have included quality labels in the time series of all the GloLakes products in revision. This helps a user to interpret rapidly from the meta-data how the data were generated. The quality labels are:**

> *"Q1: absolute volume estimated using geostatistical model and satellite-derived lake extents*
>
> *Q2: absolute volume estimated based on the V-H relationship*
>
> *Q3: relative volume estimated based on both satellite-derived heights and extents.*
>
> *Q4: relative volume estimated based on heights obtained from satellite measurements, combined with the extents of the lake derived from the area-height (A-H) relationship."*

**We also marked these for different GloLakes products in the Table 4 in the revised manuscript below:**

**Table 4** Overview of GloLakes Product Descriptions

| Filename | Type of volume | Satellite sources | Historical method | NRT method | The number of measured lakes | Time Coverage |
|---|---|---|---|---|---|---|
| Global_Lake_Absolute_Storage_LandsatPlusGREALM (1984-present).nc | Absolute | GSWD (Landsat) + G-REALM | Q1 | Q2 | 129 | 1984-current |
| Global_Lake_Absolute_Storage_LandsatPlusICESat2 (1984-present).nc | Absolute | GSWD (Landsat) + ICESat-2 | Q1 | Q2 | 24,865 | 1984-current |
| Global_Lake_Absolute_Storage_LandsatPlusSentinel2 (1984-present).nc | Absolute | GSWD (Landsat) + BLUEDOT (Sentinel-2) | Q1 | Q1 | 4,054 | 1984-current |
| Global_Lake_Relative_Storage_LandsatPlusGREALM (1993-present).nc | Relative | GSWD (Landsat) + G-REALM | Q3 | Q4 | 227 | 1993-current |
| Global_Lake_Relative_Storage_LandsatPlusICESat2 (2018-present).nc | Relative | GSWD (Landsat) + ICESat-2 | Q3 | Q4 | 24,990 | 2018-current |
| Global_Lake_Relative_Storage_Sentinel2PlusICESat2 (2018-present).nc | Relative | BLUEDOT (Sentinel-2) + ICESat-2 | Q3 | Q3 | 2740 | 2018-current |

R1C4) 3. Lack of discussion of revisit times

My apologies if I've missed it, but it seems to me this paper is entirely missing discussion about the frequency of observations, particularly for the NRT data. Thorough discussion and presentation of results about revisit times and how often you actually get an NRT observation of each lake is absolutely vital for a reader of this paper to determine the usability/relevance of the resulting dataset. As noted above, this lack of discussion of revisit times in the results/discussion section is especially important given the huge difference between ICESat-2 and Sentinel-2 revisits. For example, Figure 3 and the section about 'how many lakes can be monitored' using this approach is not useful without some discussion around frequency of observation.

**We have highlighted the revisit times in Table 1 and discussed it in the Section 3.5 Current limitations and future opportunities. However, to further clarify the difference in 'revisit' time of different datasets used in this study and their remote sensing sources and to address the reviewer's concern, we modified the paragraph in L503-528 in the revised manuscript as follows:**

*"3.5 Current limitations and future opportunities*

*Remote sensing provides an opportunity to measure water in most lakes worldwide and provide NRT or up-to-date information, which is impossible with the current in situ network. Topex/Poseidon (1992-2002), Jason 1/2/3 (2002-present), and Sentinel-6 (a.k.a. Jason-CS; 2020-present) are all able to measure lake height every ten days, which should be adequate to monitor dynamics in many lakes. Sentinel-3A/B (2016-present) has a revisit time of 27 days, still providing valuable, quasi- monthly updates on changes on water level. The dynamic estimates of lake height from these radar altimeters were seamlessly processed and derived from G-REALM. However, radar altimeters cannot detect many smaller lakes in between the sparse ground tracks. The ICESat-2 laser altimeter covers many more lakes globally, benefiting from its dense reference tracks enhanced by the six laser beams onboard. The trade-off is its temporal resolution of ~91 days, but this is still sufficient to observe seasonal changes in many lakes worldwide, and more frequent water extent mapping can be used to interpolate between these observations. This study used the Landsat-derived GSWD to estimate lake surface water extents. The main limitation of using GSWD is that it only produced monthly observations from 1984-2020 and cannot provide NRT or up-to-date information, despite its use of Landsat observations with a temporal frequency of 16 days. The GSWD are expected to be updated annually by the Joint Research Centre of the European Commission (https://global-surface-water.appspot.com/download), at which point lake extent estimates can be extended beyond 2020. In this study, we harnessed the capabilities of BLUEDOT (Sentinel-2) to expand lake surface water extent estimates beyond 2020, benefitting from the 5-day revisit interval of Sentinel-2 within the BLUEDOT framework. However, lake area data from BLUEDOT are presently available only for a subset of several thousand lakes across the globe. This issue could be overcome using MODIS, VIIRS or Sentinel-2 data directly, for example. The daily or 8-day composite MODIS and VIIRS products have a better chance to provide valid observations, but the hundred-meter range resolution is often not sufficient to accurately detect lake area changes. 5-day, 10-m resolution water extent derived from Sentinel-2 should be a promising candidate for NRT global lake monitoring. The sheer volume of data presented us with a challenge for data storage and processing, but there is no fundamental limitation that would prevent a similar approach from measuring all 170,957 lakes measured by GSWD (Landsat) in this study. The advantage of Sentinel-2 would be that it can provide NRT lake observations with low latency. The high computation and storage demands could potentially be met by cloud platforms like Google Earth Engine (GEE). In future research, we hope to consider such approaches to improve our data set."*

Specific Comments:

R1C5) Line 49-53: You might considering referencing/discussing GeoDAR here, a new global reservoir and dam database that is arguably better than the databases discussed here, in this paragraph: https://essd.copernicus.org/articles/14/1869/2022/

**Thank you. We included this reference and added one sentence (L52-54) about it in the revised manuscript.**

> "The more recent Georeferenced global Dams And Reservoirs (GeoDAR) dataset (Wang et al., 2022) not only provides the locations of 22,560 dams but also delineates the boundaries of 21,515 reservoirs around the world"

**[1] Wang, J., Walter, B. A., Yao, F., Song, C., Ding, M., Maroof, A. S., Zhu, J., Fan, C., McAlister, J. M., Sikder, S., Sheng, Y., Allen, G. H., Crétaux, J.-F., and Wada, Y.: GeoDAR: georeferenced global dams and reservoirs dataset for bridging attributes and geolocations, Earth Syst. Sci. Data, 14, 1869–1899, https://doi.org/10.5194/essd-14-1869-2022, 2022.**

R1C6) Line 74: There's a mistake here – the sentence just reads "Compared to…" but presumably there should be more text there

**Apologies for this mistake. We deleted "Compared to" here in the revised manuscript.**

R1C7) Line 81: The phrasing "whose number soars exponentially as smaller lake sizes are considered" doesn't make sense here, maybe replace with something like "which are exponentially more numerous than large lakes"

**Thank you for your suggestion. We rephased it as "which are exponentially more numerous than large lakes" in L82-83 in the revised manuscript.**

R1C8) Appreciate the adding of discussion around the different ICESat-2 products.

**Thank you again for referring us to the ICESat-2 quick look data.**

R1C9) Figure 3 (and paragraph above): When discussing what percent of lakes in a given basin that can be observed using these techniques, it is imperative to note the percent of what (i.e. what is the denominator. There are millions upon millions of lakes globally, so I assume the denominator here is the 170,000 lakes whose storage dynamics were tracked, but you should be explicit about this.

**Yes, the denominator used to calculate the percentages in Fig3b-d is the total number (170,611) of lakes whose storage dynamics have been estimated in this study (Fig3a). We clarified this in the figure caption and corresponding sentences in the revised manuscript.**

R1C10) Line 332: It's meaningless to state that 'Landsat and ICESat-2 together could measure lake water storage in nearly all rivers basins worldwide'. If ICESat-2 is only measuring a handful of lakes in a basin, is that really estimating its storage? Perhaps rephrase this sentence.

**Apologies for the confusion. We rephased this sentence to clarify what we mean here:**

> *"GSWD (Landsat) and ICESat-2 together could measure lake change in nearly all rivers basins (i.e., 234 out of 292) worldwide (Fig. 3b)."*

R1C11) Line 332-333: When you state that extents and levels are correlated for ~1/4 of all lakes, is that because ICESat-2 only observes 1/4 of all lakes, or are there only good correlations for 1/4?

**Following the change in R1C10, we modified this sentence in L372-373 in the revised manuscript to clarify this:**

> *"Extents derived from GSWD (Landsat) data and levels derived from ICESat-2 exhibit a substantial correlation in over 25% of lakes within 145 out of the total 224 river basins."*

R1C12) Line 325-330: I am confused here by the three complementary approaches – I think this should read (1) geostatistical model + extent (Landsat + geostatistical model + Sentinel-2), (2) geostatistical model + height (Landsat + geostatistical model + ICESat-2) and (3) extent + height (Sentinel-2 + Landsat). As written, it is confusing because you are essentially equating Landsat with the geostatistical model. Throughout the paper (for example, in Figure 3), please fix this and make it clear that when you say 'Landsat' you mean the GSWO dataset + the geostatistical model.

**Thank you. We changed this sentence following your suggestions:**

> *"(1) geostatistical model + extent (GSWD (Landsat) + geostatistical model + BLUEDOT (Sentinel-2)), (2) geostatistical model + height (GSWD (Landsat) + geostatistical model + V-H relationship + ICESat-2) and (3) extent + height (BLUEDOT (Sentinel-2) + ICESat-2)"*

R1C13) Line 359: What is GL?

**GL is gigalitre ($10^6$ m$^3$) and numerically equal to millions of cubic meters (MCM). We would argue GL and MCM are both commonly used units for surface water volume but generally S.I. units are preferred by scientific journals. To clarify, we added "(V in gigalitre (GL), millions of cubic meters (MCM) or $10^6$ m$^3$)" in L233-234 in the revised manuscript.**

R1C14) Figure 5. What is the unit on the y axis?

**The unit used throughout this manuscript is GL. We added the unit (GL) in this figure caption in the revised manuscript.**

R1C15) Figure 6. What is GL?

**Please see our response to R1C13.**

R1C16) Line 375: I'm not sure I agree that the relative agreement is more important than the absolute error. I can see what the authors mean here, but when discussing the comparison with the Tortini data, the authors should include the SMAPE/MAE to provide better validation context.

**This discussion pertains to the outcomes presented in Figure 4, 5 and 6, where we provided both R and SMAPE validation results, but our primary emphasis is on R values due to their importance. In Figure 7 and 8, We compared our relative change estimates in lake volume with the Tortini data, making the R value the key metric for reflecting relative agreement.**

R1C17) Figure 8: All lakes chosen to display in Figure 8 have very high R values, but there clearly are lakes with far worse agreement. This combined with the fact that the authors do not report any absolute error and only the R values (and then don't show any of the worse R values) makes me somewhat question the accuracy of the results.

**The validation results mostly show strong R values (Fig. 7), but we have included three instances of worsened results in Figure 8 in the revised manuscript.**

R1C18) Line 447: This is wrong to state that 'Landsat cannot provide NRT observations'. Landsat data IS available NRT (at 16-day revisit), but rather the JRC-GSWD product is not NRT.

**We corrected this sentence in the revised manuscript:**
> *"The main limitation of using GSWD is that it only produced monthly observations from 1984-2020 and cannot provide NRT or up-to-date information, despite its use of Landsat observations with a temporal frequency of 16 days."*

R1C19) Line 466: I'm not sure it's correct to state that the spatial resolution of SWOT is worse than Landsat and Sentinel-2. SWOT doesn't have a spatial resolution in the same way that Landsat and Sentinel-2 do (i.e. it requires spatial averaging to produce height observations). SWOT will be able to measure lake height and area ~6 million lakes globally, which is far more than the number of lakes observed here, so perhaps just remove this part about the spatial resolution of SWOT.

**Thank you for your suggestion. To avoid the confusion, we removed this sentence in the revised manuscript.**

**Response to Reviewer #2 Comments:**

I comment that the authors paid attention to my comments. In particular, they added additional details and validation results to improve the clarity and quality of this manuscript. I still have a few major concerns that some statements seem to be inaccurate or potentially biased and that some validation analysis is not convincing or completed.

**We want to thank the reviewer again for their comments and suggestions. We tried our best and added further validation of satellite-derived lake extent and height estimates in the revised manuscript. We hope that these address the reiewer's concerns. Please see our detailed responses below.**

R2C1) I have a major concern on the validation of estimated areas. Previously, I commented the uncertainty of areas due to using a global model and the global model may not be sufficient to track water area changes in each lake locally due to differences in water quality, surrounding land cover types, and atmospheric conditions (e.g., aerosol). Because of this challenge, there are multiple water indices that have been developed to track water area changes under different scenarios in recent literature. The authors' validation has little contribution in reducing my concern. First, Zhao and Gao (2018) used the same data source (Pekel et al). Comparing with Zhao and Gao is barely a comparison rather than a validation. Donchyts et al. also solely depend on one water index NDWI, while the limitations of NDWI have been acknowledged by many studies. I recall at least a couple of regional-to-global-scale studies on generating lake area time series using more sophisticated approaches (e.g., multiple water indices) rather than a global classification model. I expect the authors to be familiar with recent literature when producing a global dataset in this area. Second, validating against the two studies shows a remarkable difference. This also emphasizes my surmise of the uncertainty of estimated areas. Third, the authors did not response to my concern on the fidelity of water area changes for each lake individually. This is particularly important as a dataset paper, the users may more likely choose to use one lake for a case study than applying the dataset globally. What's the mean bias of each generated lake area time series? In sum, I strongly recommend the authors consider more quality-assured data sources and methods for the validation rather than seemingly flawed validation analysis.

**We agree that there are limitations and challenges when using remote sensing to precisely estimate area change for each lake, especially in global studies. However, it is important to clarify that our study does not focus on enhancing the current classification algorithm for estimating lake extent. Instead, our primary objective is to ensure that our lake extent estimation attains a level of proficiency comparable to other published datasets (e.g., Zhao and Gao (2018) and Pekel et al. (2016)). Then, we can gain the confidence to convert extent to volume using a geo-statistical model. Second, we argue that the GSWD data used in this study is a reliable global surface water classification product as it used non-parametric classifiers (expert systems), rather than single or multiple indices, to account for uncertainties in the data. The thorough validation of this classification algorithm can be found in Pekel et al. (2016). Third, relatively larger bias metrics does not necessarily imply that the estimates contain significant errors, as different datasets may have different definitions of lake boundaries, e.g. one can include broader upstream channels and ephemeral connected water bodies as part of lake area. For example, lake boundaries from HydroLAKES and Donchyts et al. (2022) are different to some extent in many lakes but**

conceptually and technically equally justifiable. Lastly, we do not use all lake area time series derived from GSWD. Instead, we only used GSWD-derived lake area time series, which are significantly correlated with other data sources, i.e., Bluedot (Sentinel-2) lake area time series, or GREALM/ICESat-2 lake height time series (L240-243 and Figure 3). We assumed that if two satellites source show consistency in lake changes, the derived lake area and height time series are more likely to be reliable.

Incorporating your suggestions, we expanded our analysis beyond the area time series comparison of 16,419 lakes against Zhao and Gao (2018) and Donchyts et al. (2022). To further validate our lake area estimates, we conducted a comparison with Sentinel-2 data, utilizing the capabilities of Google Earth Engine. The revised manuscript includes an additional paragraph, a figure, and a table to thoroughly discuss these new validation results:

> *"To further assess our lake extent estimates, we manually derived lake area based on bands 8 (near infrared), 4 (red) and 3 (green) from high-resolution Sentinel-2 imagery using Google Earth Engine and compared it against lake area derived in this study (Figure S2 and Table S1). We selected two Sentinel-2 images without cloud cover - one with a relatively small extent and another with a relatively large extent - for each of 20 lakes chosen to represent different continents and climates and with varying surface water color, lake shape and size and surrounding land cover. We delineated the lake area using Sentinel-2 surface reflectances in the three bands and visually validated it with a false-color image. The Sentinel-2 derived lake area was then compared to GSWD (Landsat) derived lake area. The results show that the average difference of lake area estimates derived between GSWD (Landsat) and Sentinel-2 for all 40 pairs is 2.62% (Table S1). The estimated lake area differences do not vary systematically with geographical location, surrounding land cover or lake shape and size. Most importantly, the area estimates for both small and large extents exhibited very strong consistency between the two satellite sources (Figure S2). This confirms GSWD (Landsat) can successfully capture changes in lake area. Comparatively large biases (from 3-10%) are observed exclusively in lakes with surrounding ephemeral disconnected surface water bodies or emerging islands within the lake during dry periods, such as Lake Pozuelos in Argentina, Lake Baia Grande in Brazil, and Lake Aksehir in Turkey (Figure S2 and Table S1)."*

[Figure]

**Figure S2.** Comparisons between GSWD lake extent binary map (yellow: wet pixels; blue: dry pixels) and Sentinle-2 false color imagery (yellow line: derived lake extent boundaries) at different time for selected 20 lakes around the world.

[Figure]

**Figure S2 (continued).** Comparisons between GSWD lake extent binary map (yellow: wet pixels; blue: dry pixels) and Sentinel-2 false color imagery (yellow line: derived lake extent boundaries) at different time for selected 20 lakes around the world.

[Figure]

**Figure S2 (continued).** Comparisons between GSWD lake extent binary map (yellow: wet pixels; blue: dry pixels) and Sentinel-2 false color imagery (yellow line: derived lake extent boundaries) at different time for selected 20 lakes around the world.

[Figure]

**Figure S2 (continued).** Comparisons between GSWD lake extent binary map (yellow: wet pixels; blue: dry pixels) and Sentinel-2 false color imagery (yellow line: derived lake extent boundaries) at different time for selected 20 lakes around the world.

**Table S1** Comparison results of lake extent estimates derived between Landsat-GSWD and Sentinel-2 (statistics results from Figure S2)

| ID | Lake Name | Country | Latitude | Longitude | Date | Landsat-GSWD lake extent (km²) | Sentinel-2 lake extent (km²) | Difference (%) |
|---|---|---|---|---|---|---|---|---|
| 634 | Petit Manicouagan | Canada | 51.82 | -67.80 | 2020-06 | 312 | 313 | 0.32 |
| 634 | Petit Manicouagan | Canada | 51.82 | -67.80 | 2019-07 | 305 | 320 | 4.69 |
| 448 | | Canada | 58.28 | -96.99 | 2019-07 | 226 | 227 | 0.44 |
| 448 | | Canada | 58.28 | -96.99 | 2021-08 | 229 | 224 | 2.23 |
| 9291 | Millerton Lake | United States of America | 37.00 | -119.70 | 2020-02 | 17.3 | 17.5 | 1.14 |
| 9291 | Millerton Lake | United States of America | 37.00 | -119.70 | 2020-09 | 15.2 | 14.9 | 2.01 |
| 16140 | Gove | Angola | -13.45 | 15.87 | 2021-02 | 101 | 102 | 0.98 |
| 16140 | Gove | Angola | -13.45 | 15.87 | 2019-04 | 76 | 78 | 2.56 |
| 15699 | | India | 17.23 | 79.52 | 2019-02 | 10.7 | 11.3 | 5.31 |
| 15699 | | India | 17.23 | 79.52 | 2019-04 | 2.92 | 3 | 2.67 |
| 173266 | | Uzbekistan | 38.87 | 66.42 | 2019-09 | 1.84 | 1.83 | 0.55 |
| 173266 | | Uzbekistan | 38.87 | 66.42 | 2019-05 | 2.7 | 2.8 | 3.57 |
| 16573 | Burrendong | Australia | -32.67 | 149.11 | 2019-04 | 16.8 | 17 | 1.18 |
| 16573 | Burrendong | Australia | -32.67 | 149.11 | 2021-02 | 45.4 | 45.2 | 0.44 |
| 1564 | Abayata | Ethiopia | 7.61 | 38.61 | 2021-01 | 150 | 149 | 0.67 |
| 1564 | Abayata | Ethiopia | 7.61 | 38.61 | 2019-01 | 71.2 | 70.5 | 0.99 |
| 173736 | Vadomojon | Spain | 37.64 | -4.23 | 2019-08 | 4.97 | 5.06 | 1.78 |
| 173736 | Vadomojon | Spain | 37.64 | -4.23 | 2019-02 | 6.93 | 7.19 | 3.62 |
| 10338 | Pozuelos | Argentina | -22.32 | -65.99 | 2019-05 | 95 | 98.6 | 3.65 |
| 10338 | Pozuelos | Argentina | -22.32 | -65.99 | 2019-11 | 31.4 | 33.3 | 5.71 |
| 10255 | Baia Grande | Brazil | -15.53 | -60.19 | 2020-03 | 56.6 | 51.7 | 9.48 |
| 10255 | Baia Grande | Brazil | -15.53 | -60.19 | 2021-06 | 43.7 | 40.3 | 8.44 |
| 1335 | Aksehir | Turkey | 38.51 | 31.42 | 2019-07 | 91 | 85.5 | 6.43 |
| 1335 | Aksehir | Turkey | 38.51 | 31.42 | 2020-07 | 96 | 92 | 4.35 |
| 11579 | | Russia | 67.76 | 124.22 | 2021-06 | 191.8 | 189 | 1.48 |
| 11579 | | Russia | 67.76 | 124.22 | 2020-08 | 186 | 183 | 1.64 |
| 15448 | | China | 26.71 | 117.12 | 2019-09 | 31.8 | 33.1 | 3.93 |
| 15448 | | China | 26.71 | 117.12 | 2021-01 | 31.6 | 32 | 1.25 |
| 1092 | Flasjon | Sweden | 64.14 | 15.91 | 2020-05 | 254 | 252 | 0.79 |
| 1092 | Flasjon | Sweden | 64.14 | 15.91 | 2021-07 | 254 | 253 | 0.40 |
| 9372 | Lake Altus | United States of America | 34.89 | -99.29 | 2019-03 | 22.9 | 22.1 | 3.62 |
| 9372 | Lake Altus | United States of America | 34.89 | -99.29 | 2021-06 | 15.9 | 16 | 0.62 |
| 15723 | | Mali | 15.78 | -4.53 | 2019-03 | 36.4 | 36.1 | 0.83 |
| 15723 | | Mali | 15.78 | -4.53 | 2019-09 | 28.9 | 29.1 | 0.69 |
| 161113 | | Russia | 56.00 | 28.71 | 2019-04 | 10.5 | 10.6 | 0.94 |
| 161113 | | Russia | 56.00 | 28.71 | 2021-06 | 9.3 | 9.5 | 2.11 |
| 13689 | | Kazakhstan | 51.37 | 61.86 | 2021-05 | 30.6 | 29.2 | 4.79 |
| 13689 | | Kazakhstan | 51.37 | 61.86 | 2021-09 | 0 | 0 | 0.00 |

| 10165 | Brazil | -10.23 | -44.68 | 2021-07 | 3.4 | 3.2 | 6.25 |
| 10165 | Brazil | -10.23 | -44.68 | 2019-09 | 0.93 | 0.91 | 2.20 |

R2C2) I am still concerned about the accuracy of the produced datasets in cold regions given cold regions have a large share of Earth's lakes. The authors claimed that they could not find any in-situ data in Canada. I am not sure if the authors really tried the best to find the in-situ data in Canada. Here is in-situ data for Canadian lakes: https://wateroffice.ec.gc.ca/. Some countries in Northern Europe may have in-situ data as well. In many lakes, storage changes are dominated by level changes (Cooley et al. 2021 Nature). As the authors used different approaches to derive levels in order to estimate the storage time series, what's the mean level error of each generated time series? How does the level error vary among different approaches applied? I would like to remind the authors that in-situ level data has a much larger spatial coverage than 438 lakes used in the current validation, particularly for small lakes. How many natural lakes are included in the validation?.

**Thank you for your suggestions and sharing the link to these in situ data. In the revised manuscript, we included validation results on lake area estimates for 10 lakes in cold regions, including two lakes in Canada, one lake in Sweden, and two lakes in Russia. The validation results show strong agreement in lake area estimates between our study and GEE Sentinel-2 manually derived data in cold regions. Please see the detailed response to R2C1. Secondly, while we have successfully estimated lake extents and heights for the majority of lakes in cold regions, our analysis does not encompass all lakes in these regions. Rather, we have specifically focused on providing data for lakes that exhibit consistent satellite-derived extents and heights over time, ensuring the reliability of our published estimates through significant correlation. This rigorous selection process has led us to present data for less than 10% of the lakes in cold regions (refer to Figure 3b-d), as we prioritize data accuracy and precision. Lastly, we agree that there are more in situ level data available from different countries, but obtaining data from most of these agencies poses a significant challenge due to the lack of easily accessible means for discovering and downloading the information. This has been a challenge to many global studies. For example, the global lake study of Cooley et al. (2021) only used in-situ level data from the USA for validating lake height estimates (Figure S3).**

**We tried our best to collect more in situ level data and used them to validate our lake height estimates. We were unable to locate any in situ lake level data from Donchyts et al. (2022) that could be utilized to validate our lake level estimates, but we did use their in situ lake extent data to validate our extent estimates for several hundred lakes (most of their data have in situ lake extent and volume data, but not necessarily level data). We invested time and effort in understanding the API for accessing in situ data from Environment Canada. Additionally, we submitted a request to the Australian Bureau of Meteorology to obtain their in situ data. Overall, in addition to the validation results in the USA from Cooley et al. (2021), we included lake height validation results for two more countries (one in a cold region) in the revised manuscript (Figure 9):**

> *"To validate satellite-derived lake heights used to estimate NRT storages in this study, we obtained in situ lake level measurements from the Australian Bureau of Meteorology (http://www.bom.gov.au/water/index.shtml) and Environment Canada (https://wateroffice.ec.gc.ca/). In*

*total, we identified 96 lakes in Australia (including 83 reservoirs) and 54 lakes in Canada (including 23 reservoirs) with in situ level data corresponding to our study. Following the same validation approach as Cooley et al. (2021), we compared temporal changes in lake heights between in situ data and altimetry data. The results show strong agreement between ICESat-2 and in situ measurements, with a mean absolute error (MAE) of 0.23 m and a mean Pearson correlation (R) of 0.99 in Australia and 0.19 m, and 0.97 in Canada (Figure 9).''*

[Figure]

**Figure S3** Evaluation of ICESat-2 derived lake heights against in situ gauge measurements from United States Geological Survey (Cooley et al., 2021).

[Figure]

**Figure 9** Evaluation of ICESat-2 derived lake heights against in situ gauge measurements from Australian Bureau of Meteorology (BoM) and Environment Canada (EC).

---

## Author Response (AR4)

**Response to Reviewers**

**''GloLakes: water storage dynamics for 27,000 lakes globally from 1984 to present derived from satellite altimetry and optical imaging'' by Jiawei Hou et al.**

**We express our gratitude to the editor and two reviewers for generously dedicating their time and substantial effort to assess our manuscript. We have carefully considered all comments and suggestions provided by the editor and reviewers and have made modifications accordingly (review comments in blue, our response in black bold font).**

**Response to Editor's Comments:**

Many thanks for your thorough revisions of your manuscript, which has now been seen by the referees. Both referees acknowledge your improvements of the manuscript and I am happy to accept the paper for publication after some final minor adjustments. While the suggestions of referee #1, should be straight forward to address - I also want to put some of the critique of the second referee into context. Here I find that a careful adjustment of the terminology may alleviate some misunderstandings regarding your analysis. In particular I would suggest that you consider the following points:

**We extend our gratitude to the editor for the encouragement and positive feedback. The insightful suggestions provided below were valuable in enhancing the quality of this manuscript.**

EC1) 1. Clarify how remote sensing data are processed in the abstract (and possibly title) to avoid confusion (i.e. that lake volumes are estimated based on topographic characteristics and lake properties that can be observed by remote sensing).

**We clarified this in the abstract in the revised manuscript:**

> *"Measuring the spatiotemporal dynamics of lake and reservoir water storage is fundamental for assessing the influence of climate variability and anthropogenic activities on water quantity and quality. Previous studies estimated relative water volume changes for lakes where both satellite-derived extent and radar altimetry data are available. This approach is limited to only a few hundred lakes worldwide and cannot estimate absolute (i.e., total volume) water storage. We increased the number of measured lakes by a factor of 300 by using high-resolution Landsat and Sentinel-2 optical remote sensing and ICESat-2 laser altimetry, in addition to radar altimetry from the Topex/Poseidon, Jason-1, -2 and -3, and Sentinel-3 and -6 instruments. Historical time series (1984-2020) of water storage could be derived for more than 170,000 lakes globally with a surface area of at least 1 km2, representing 99% of the total volume of all water stored in lakes and reservoirs globally. Specifically, absolute lake volumes are estimated based on topographic characteristics and lake properties that can be observed by remote sensing. In addition to that, we also generated relative lake volume changes solely based on satellite-derived heights and extents if both available. Within this data set, we investigated how many lakes can be*

*measured in near real-time (2020-current) in basins worldwide. We developed an automated workflow for near real-time global lake monitoring of more than 27,000 lakes. The historical and near real-time lake storage dynamics data from 1984 to current are publicly available through https://doi.org/10.25914/K8ZF-6G46 (Hou et al., 2022) and a web-based data explorer www.globalwater.online."*

EC2) The term "validation" might be misunderstood by some of the readers, since it can imply comparison with an indisputable truth. I acknowledge that for many environmental variables the availability of observations or the quality of an estimate can be a hurdle. Therefore I find that any comparison to independent data is valuable. For example it may not always be possible to decide that an alternative data-product is better, but demonstrating consistency can increase confidence in new estimates. Similarly comparison using a small subset of the data to available in-situ observations at targeted locations may also be valid in light of limited data-availability. Therefore I suggest that you consider to use terms like "benchmarking", "checking consistency" or similar.

**Thank you so much for your comments and suggestions. In the revised manuscript, to avoid the misunderstanding, we changed "validation" to "benchmarking" where we compared our GloLakes database against other, independent data. We only used the term "validation" where we compared our data against in situ data, which, although still not indisputable, would likely be accepted as closer to 'truth'.**

**Response to Reviewer #1 Comments:**

I thank the authors for responding to all my comments and updating the terminology in the paper. Upon reading the revised version, I think the text does do a much better job of communicating what each dataset represents and what they can/cannot do. I just have a few remaining minor comments (see below):
Note – all lines referenced below are from the tracked changes version.

**Your valuable comments and insights have made the manuscript more effective in conveying the dataset information accurately. We're pleased that you found the revised version to be an improvement. Thank you for your support and guidance.**

R1C1) Line 114: "Explore all ready available, validated and frequently updated satellite data sources" does not accurately describe this dataset, as one could argue that Sentinel-2 is readily available, validated and updated with far more lakes than the version used here (BLUEDOT). Suggest rephrasing to say something like we test "some satellite data sources" or replace "satellite data sources" with something more specific about testing satellite surface water data sources.

**We rephrased this sentence in the revised manuscript:**

> *"The underlying strategy in developing this global lake monitoring system was to evaluate a selection of readily available, validated, and frequently updated satellite data sources, explore the relative advantages of each selected source and combine them to complement their respective weaknesses."*

R1C2) Line 376: Replace "substantial" with significant. I assume you mean significant at p < 0.05? I'd also suggest clarifying 25% of which lakes you're referring to here – 25% of the 170,000 lakes in GSWD?

**Apologies for the confusion. This refers to the total number of lakes that measured by both satellite sources in each basin. We rephrased this sentence in the revised manuscript:**

> *"Extents derived from GSWD (Landsat) data and levels derived from ICESat-2 exhibit a significant correlation in over 25% of lakes measured by both in each of 145 river basins."*

R1C3) Line 377: Change "this feature" to "these correlations" or similar

**We changed this to "***The correlation pattern is evenly distributed across the continents...***" in the revised manuscript.**

R1C4) Line 381: Again clarify that "half of the lakes" refers to half to the 170,000 GSWD lakes.

**We rephrased this sentence in the revised manuscript:**

> *"There are 58 basins in each of which over half of the lakes measured by both satellite sources can be monitored by that method, mainly located in the USA, southeastern South America, the Mediterranean, southern Africa, southern Asia, and Australia."*

R1C5) Line 505-506: "adequate to monitor dynamics in many lakes" is not specific enough – these sensors are great for measuring very large lakes, but not useful otherwise.

**We changed this to "** ...*adequate to monitor dynamics in large lakes.***" in the revised manuscript.**

R1C6) The additional text in section 3.5 is rather poorly written/structured. The text about Sentinel-2 (lines 516-525) is a very needed addition, and I understand that the authors were trying to add additional context. However, while the information is largely here, the structure doesn't quite work, particularly the transition between the discussion of the altimeters and the Landsat-derived GSWD. I'd suggest rewriting this section and splitting it into separate discussions about the extent observations from GSWD/Landsat/BLUEDOT/Sentinel-2, followed by a discussion of the altimetry and then the nice bit at the end about the potential of radar.

**Thank you for your suggestions. We restructured this section in the revised manuscript:**

[revised manuscript text omitted]

R1C7) Nice conclusion section.

**We appreciate your positive feedback.**

**Response to Reviewer #2 Comments:**

I appreciate the efforts that the authors made in the revisions. However, the authors should really avoid misleading information in the manuscript.

**Thank you for your feedback on this manuscript. We acknowledge your concerns and would like to provide a thorough explanation of our database validation approach, as well as the limitations we encountered when validating this new global dataset. In response to the Editor's recommendations, we revised the manuscript to enhance the clarity regarding the processing of remote sensing data for our various products and the methods used for validation. Please see our detailed responses to EC1 and EC2.**

R2C1) 1. In the section 3.1 Lake area estimation validation, the authors chose to compare with the Zhao and Gao (2018) while their method for estimating water areas is almost identical to Zhao and Gao. What's the purpose of this comparison and even labeling it as validation? It is almost 100% expected that these two sets of areas are (nearly) the same. In other words, the results (label as validation) here are really misleading.

In the comparison with Donchyts et al. 2022, if the authors only focus on generating one-snap of lake area (e.g., similar to the HydroLakes database), I would be likely convinced by this type of comparison, although the authors may need to provide evidence supporting that Donchyts et al. 2022 is better than theirs and why they did not adopt the methodology of Donchyts et al. 2022. However, the authors aim to focus on water area change over time, this comparison did not provide sufficient information on the applicability of their products, particularly for estimating lake volume changes. As shown in Figure 2b, the discrepancy in area become significantly larger for lakes smaller than 10 km2, although both datasets were generated from the same source (Landsat images). This is unlikely caused by the resolution limitation of Landsat given 30-m resolution should be sufficient for monitoring water bodies larger than 1 km2 (the minimum size of studied lakes here). I would like to reiterate my suggestion that the authors should consider reporting the mean absolute error on a lake-to-lake basis and compare the error to the variability in the time series. This would be more important than the results they showed in Figure 2, in my view.

I acknowledge that the authors did additional validation on 20 lakes (two snapshots per lake) using the higher-resolution sentinel-2. If this is a local or regional scale study, this may work. For a global study of 170k lakes, this value of this evaluation on 20 lakes is really limited.

**We think the comparison against Zhao and Gao (2018) is necessary. Although we utilized the same data source (GSWD), our approach to gap-filling contaminated images differs. If these two datasets exhibit consistency, it would indicate the effectiveness of our gap-filling algorithm, at least akin to that of Zhao and Gao (2018).**

**In our opinion, the relatively large discrepancy observed between our database and Donchyts et al. 2022 does not dominate the comparison results. We have also conducted investigations into these discrepancies and determined that they are primarily attributable to differences in defining lake boundaries, at times making it challenging to conceptually distinguish lakes from interconnected water bodies (L317). Additionally, in both comparison analyses, we reported the mean absolute error as SMAPE on a lake-to-lake time series basis (L313 and L319).**

**We acknowledge the limitations of evaluating this global database. We conducted a manual comparison with high-resolution Sentinel-2 data. Although we incorporated additional benchmarking results for 20 lakes, we tried to choose a diverse set of lakes (i.e., in different continents and climate zones and with varying surface water color, lake shape and size and surrounding land cover.) for this analysis. Therefore, our comparison results could be considered representative for global studies.**

R2C2) 2. The added Figure 9 is also misleading. The authors chose to validate the water levels derived from ICESat-2 (2018 to now) using in-situ level, while their study focuses on generating lake volume time series from 1984 to the present. The authors seem to use the validated accuracy of ICESat-2 levels to report the accuracy for the derived water levels over the historical period (1984 - ). In the last review, I commented that the geospatial model may not work for estimating volume changes and suggested that additional validation on the levels estimated from the geospatial model to be validated. What a surprise to see such a misleading response? Additionally, the authors selected 10 lakes for this validation without a careful thought of whether the number is sufficient and whether the samples are representative.

**By adding Figure 9 with satellite-derived lake height validation results for 96 lakes, we believe our study now has a thorough validation/benchmarking analysis of all the lake water resources variables, i.e., height, extent, and storage. However, these validation/benchmarking analyses indeed have been limited by the availability of in situ data or published datasets.**

**We think the geo-statistical model mainly determines the lake bathymetry, rather than volume changes. The accuracy of volume changes mainly relies on the precision of lake height and extent measurements, both of which have undergone validation and benchmarking within this study. In the manuscript, we reported both correlation and bias error for lake height and extent measurements and the geo-statistical model used in this study (L458-459, L313-319 and L399). As there are no available in situ lake extent data, we compared satellite extent measurements against published datasets. While satellite-derived lake heights and the geo-statistical model have been validated by in situ data where available.**

R2C3) 3. The authors should carefully go through the entire manuscript to eliminate possible confusions. Many places are misleading. For example, the title implies that the lake volume derived using satellite altimetry and optical imaging. But the generated data largely came from the geostatistical model that seems to be empirical (definitely not based on satellites). In the Methods, "relative storage products are unaffected by any bias from the geostatistical model". I think that volume changes estimated from empirical models are subject to large uncertainties. The authors should clarify that what this statement means..

**In this study, we not only generated absolute lake storage (which is based on the geo-statistical model), but also relative lake storage (relying solely on satellite-derived lake height and extent measurements). The relative lake storage time series were not estimated by the geo-statistical model, therefore we claim that relative storage products are unaffected by any bias from the geostatistical model. We labeled our different products with corresponding methods (L549-553). In the revised manuscript, we clarified this in the abstract.**